# SGD Through the Lens of Kolmogorov Complexity

## Abstract

We initiate a thorough study of the dynamics of stochastic gradient descent (SGD) under minimal assumptions using the tools of entropy compression. Specifically, we characterize a quantity of interest which we refer to as the *accuracy discrepancy*. Roughly speaking, this measures the average discrepancy between the model accuracy on batches and large subsets of the entire dataset. We show that if this quantity is sufficiently large, then SGD finds a model which achieves perfect accuracy on the data in $O(1)$ epochs. On the contrary, if the model cannot perfectly fit the data, this quantity must remain below a *global* threshold, which only depends on the size of the dataset and batch.

We use the above framework to lower bound the amount of randomness required to allow (non-stochastic) gradient descent to escape from local minima using perturbations. We show that even if the model is *extremely overparameterized*, at least a linear (in the size of the dataset) number of random bits are required to guarantee that GD escapes local minima in subexponential time.

## 1 Introduction

Stochastic gradient descent (SGD) is at the heart of modern machine learning. However, we are still lacking a theoretical framework that explains its performance for general, non-convex functions. Current results make significant assumptions regarding the model. Global convergence guarantees only hold under specific architectures, activation units, and when models are extremely overparameterized (Du et al., 2019; Allen-Zhu et al., 2019; Zou et al., 2018; Zou and Gu, 2019). In this paper, we take a step back and explore what can be said about SGD under the most *minimal* assumptions. We only assume that the loss function is differentiable and $L$-smooth, the learning rate is sufficiently small and that models are initialized randomly. Clearly, we cannot prove general convergence to a global minimum under these assumptions. However, we can try and understand the *dynamics* of SGD - what types of execution patterns can and cannot happen.

**Motivating example**: Suppose hypothetically, that for every batch, the accuracy of the model after the Gradient Descent (GD) step on the batch is 100%. However, its accuracy on the set of previously seen batches (including the current batch) remains at 80%. Can this process go on forever? At first glance, this might seem like a possible scenario. However, we show that this cannot be the case. That is, if the above scenario repeats sufficiently often the model must eventually achieve 100% accuracy on the *entire dataset*.

To show the above, we identify a quantity of interest which we call the *accuracy discrepancy* (formally defined in Section 3). Roughly speaking, this is how much the model accuracy on a batch differs from the model accuracy on all previous batches in the epoch. We show that when this quantity (averaged over epochs) is higher than a certain threshold, we can guarantee that SGD convergence to 100% accuracy on the dataset within $O(1)$ epochs w.h.p[1]. We note that this threshold is *global*, that is, it only depends on the size of the dataset and the size of the batch. In doing so, we provide a *sufficient condition* for SGD convergence.

The above result is especially interesting when applied to weak models that cannot achieve perfect accuracy on the data. Imagine a dataset of size $n$ with random labels, a model with $n^{0.99}$ parameters, and a batch of size $\log n$. The above implies that the accuracy discrepancy must eventually go below

---

[1] With high probability means a probability of at least $1 - 1/n$, where $n$ is the size of the dataset.

the global threshold. In other words, the model cannot consistently make significant progress on batches. This is surprising because even though the model is underparameterized with respect to the entire dataset, it is extremely overparameterized with respect to the batch. We verify this observation experimentally (Appendix B). This holds for a single GD step, but what if we were to allow many GD steps per batch, would this mean that we still cannot make significant progress on the batch? This leads us to consider the role of randomness in (non-stochastic) gradient descent.

It is well known that overparameterized models trained using SGD can perfectly fit datasets with random labels (Zhang et al., 2017). It is also known that when models are sufficiently overparameterized (and wide) GD with *random initialization* convergences to a near global minimum (Du et al., 2019). This leads to an interesting question: how much randomness does GD require to escape local minima efficiently (in polynomial time)? It is obvious that without randomness we could initialize GD next to a local minimum, and it will never escape it. However, what about the case where we are provided an adversarial input and we can *perturb* that input (for example, by adding a random vector to it), how many bits of randomness are required to guarantee that after the perturbation GD achieves good accuracy on the input in polynomial time?

In Section 4 we show that if the amount of randomness is sublinear in the size of the dataset, then for any differentiable and $L$-smooth model class (e.g., a neural network architecture), there are datasets that require an exponential running time to achieve any non-trivial accuracy (i.e., better than $1/2 + o(1)$ for a two-class classification task), even if the model is extremely overparameterized. This result highlights the importance of randomness for the convergence of gradient methods. Specifically, it provides an indication of why SGD converges in certain situations and GD does not. We hope this result opens the door to the design of randomness in other versions of GD.

**Outline of our techniques**     We consider batch SGD, where the dataset is shuffled once at the beginning of each epoch and then divided into batches. We do not deal with the generalization abilities of the model. Thus, the dataset is always the training set. In each epoch, the algorithm goes over the batches one by one, and performs gradient descent to update the model. This is the "vanilla" version of SGD, without any acceleration or regularization (for a formal definition, see Section 2). For the sake of analysis, we add a termination condition after every GD step: if the accuracy on the entire dataset is 100% we terminate. Thus, in our case, termination implies 100% accuracy.

To achieve our results, we make use of *entropy compression*, first considered by Moser and Tardos (2010) to prove a constructive version of the Lovász local lemma. Roughly speaking, the entropy compression argument allows one to bound the running time of a randomized algorithm[2] by leveraging the fact that a random string of bits (the randomness used by the algorithm) is computationally incompressible (has high Kolmogorov complexity) w.h.p. If one can show that throughout the execution of the algorithm, it (implicitly) compresses the randomness it uses, then one can bound the number of iterations the algorithm may execute without terminating. To show that the algorithm has such a property, one would usually consider the algorithm after executing $t$ iterations, and would try to show that just by looking at an "execution log" of the algorithm and some set of "hints", whose size together is considerably smaller than the number of random bits used by the algorithm, it is possible to reconstruct all of the random bits used by the algorithm.

We apply this approach to SGD with an added termination condition when the accuracy over the entire dataset is 100%. Thus, termination in our case guarantees perfect accuracy. The randomness we compress is the bits required to represent the random permutation of the data at every epoch. So indeed the longer SGD executes, the more random bits are generated. We show that under our assumptions it is possible to reconstruct these bits efficiently starting from the dataset $X$ and the model after executing $t$ epochs. The first step in allowing us to reconstruct the random bits of the permutation in each epoch is to show that under the $L$-smoothness assumption and a sufficiently small step size, SGD is *reversible*. That is, if we are given a model $W_{i+1}$ and a batch $B_i$ such that $W_{i+1}$ results from taking a gradient step with model $W_i$ where the loss is calculated with respect to $B_i$, then we can *uniquely* retrieve $W_i$ using only $B_i$ and $W_{i+1}$. This means that if we can efficiently encode the batches used in every epoch (i.e., using less bits than encoding the entire permutation of the data), we can also retrieve all intermediate models in that epoch (at no additional cost). We prove this claim in Section 2.

---

[2] We require that the number of the random bits used is proportional to the execution time of the algorithm. That is, the algorithm flips coins for every iteration of a loop, rather than just a constant number at the beginning of the execution.

The crux of this paper is to show that when the accuracy discrepancy is high for a certain epoch, the batches can indeed be compressed. To exemplify our techniques let us consider the scenario where, in every epoch, just after a single GD step on a batch we consistently achieve perfect accuracy on the batch. Let us consider some epoch of our execution, assume we have access to $X$, and let $W_f$ be the model at the end of the epoch. If the algorithm did not terminate, then $W_f$ has accuracy at most $1 - \epsilon$ on the entire dataset (assume for simplicity that $\epsilon$ is a constant). Our goal is to retrieve the last batch of the epoch, $B_f \subset X$ (without knowing the permutation of the data for the epoch). A naive approach would be to simply encode the indices in $X$ of the elements in the batch. However, we can use $W_f$ to achieve a more efficient encoding. Specifically, we know that $W_f$ achieves 1.0 accuracy on $B_f$ but only $1 - \epsilon$ accuracy on $X$. Thus it is sufficient to encode the elements of $B_f$ using a smaller subset of $X$ (the elements classified correctly by $W_f$, which has size at most $(1 - \epsilon) |X|$). This allows us to significantly compress $B_f$. Next, we can use $B_f$ and $W_f$ together with the reversibility of SGD to retrieve $W_{f-1}$. We can now repeat the above argument to compress $B_{f-1}$ and so on, until we are able to reconstruct all of the random bits used to generate the permutation of $X$ in the epoch. This will result in a linear reduction in the number of bits required for the encoding.

In our analysis, we show a generalized version of the scenario above. We show that high accuracy discrepancy implies that entropy compression occurs. For our second result, we consider a modified SGD algorithm that instead of performing a single GD step per batch, first perturbs the batch with a limited amount of randomness and then performs GD until a desired accuracy on the batch is reached. We assume towards contradiction that GD can always reach the desired accuracy on the batch in subexponential time. This forces the accuracy discrepancy to be high, which guarantees that we always find a model with good accuracy. Applying this reasoning to models of sublinear size and data with random labels we arrive at a contradiction, as such models cannot achieve good accuracy on the data. This implies that when we limit the amount of randomness GD can use for perturbations, there must exist instances where GD requires *exponential* running time to achieve good accuracy.

**Related work**     There has been a long line of research proving convergence bounds for SGD under various simplifying assumptions such as: linear networks (Arora et al., 2019; 2018), shallow networks (Safran and Shamir, 2018; Du and Lee, 2018; Oymak and Soltanolkotabi, 2019), etc. However, the most general results are the ones dealing with deep, overparameterized networks (Du et al., 2019; Allen-Zhu et al., 2019; Zou et al., 2018; Zou and Gu, 2019). All of these works make use of NTK (Neural Tangent Kernel)(Jacot et al., 2018) and show global convergence guarantees for SGD when the hidden layers have width at least $poly(n, L)$ where $n$ is the size of the dataset and $L$ is the depth of the network. We note that the exponents of the polynomials are quite large.

A recent line of work by Zhang et al. (2022) notes that in many real world scenarios models do not converge to stationary points. They instead take a different approach which, similar to us, studies the dynamics of neural networks. They show that under certain assumptions (e.g., considering a fully connected architecture with sub-differentiable and coordinate-wise Lipschitz activations and weights laying on a compact set) the change in training loss gradually converges to 0, even if the full gradient norms do not vanish.

In (Du et al., 2017) it was shown that GD can take exponential time to escape saddle points, even under random initialization. They provide a highly engineered instance, while our results hold for many model classes of interest. Jin et al. (2017) show that adding perturbations *during* the executions of GD guarantees that it escapes saddle points. This is done by occasionally perturbing the parameters within a ball of radius $r$, where $r$ depends on the properties of the function to be optimized. Therefore, a *single* perturbation must require an amount of randomness *linear* in the number of parameters.

## 2    PRELIMINARIES

We consider the following optimization problem. We are given an input (dataset) of size $n$. Let us denote $X = \{x_i\}_{i=1}^{n}$ (Our inputs contain both data and labels, we do not need to distinguish them for this work). We also associate every $x \in X$ with a unique id of $\lceil \log n \rceil$ bits. We often consider batches of the input $B \subset X$. The size of the batch is denoted by $b$ (all batches have the same size). We have some model whose parameters are denoted by $W \in \mathbb{R}^d$, where $d$ is the model dimension. We aim to optimize a goal function of the following type: $f(W) = \frac{1}{n} \sum_{x \in X} f_x(W)$, where the functions $f_x : \mathbb{R}^d \to \mathbb{R}$ are completely determined by $x \in X$. We also define for every set $A \subseteq X$: $f_A(W) = \frac{1}{|A|} \sum_{x \in A} f_x(W)$. Note that $f_X = f$.

We denote by $acc(W, A) : \mathbb{R}^d \times 2^X \rightarrow [0, 1]$ the accuracy of model $W$ on the set $A \subseteq X$ (where we use $W$ to classify elements from $X$). Note that for $x \in X$ it holds that $acc(W, x)$ is a binary value indicating whether $x$ is classified correctly or not. We require that every $f_x$ is differentiable and $L$-smooth: $\forall W_1, W_2 \in \mathbb{R}^d, \|\nabla f_x(W_1) - \nabla f_x(W_2)\| \leq L\|W_1 - W_2\|$. This implies that every $f_A$ is also differentiable and $L$-smooth. To see this consider the following:

$$\|\nabla f_A(W_1) - \nabla f_A(W_2)\| = \|\frac{1}{|A|} \sum_{x \in A} \nabla f_x(W_1) - \frac{1}{|A|} \sum_{x \in A} \nabla f_x(W_2)\|$$

$$= \frac{1}{|A|}\|\sum_{x \in A} \nabla f_x(W_1) - \nabla f_x(W_2)\| \leq \frac{1}{|A|} \sum_{x \in A} \|\nabla f_x(W_1) - \nabla f_x(W_2)\| \leq L\|W_1 - W_2\|$$

We state another useful property of $f_A$:

**Lemma 2.1.** *Let $W_1, W_2 \in \mathbb{R}^d$ and $\alpha < 1/L$. For any $A \subseteq X$, if it holds that $W_1 - \alpha\nabla f_A(W_1) = W_2 - \alpha\nabla f_A(W_2)$ then $W_1 = W_2$.*

*Proof.* Rearranging the terms we get that $W_1 - W_2 = \alpha\nabla f_A(W_1) - \alpha\nabla f_A(W_2)$. Now let us consider the norm of both sides: $\|W_1 - W_2\| = \|\alpha\nabla f_A(W_1) - \alpha\nabla f_A(W_2)\| \leq \alpha \cdot L\|W_1 - W_2\| < \|W_1 - W_2\|$ Unless $W_1 = W_2$, the final strict inequality holds which leads to a contradiction. $\square$

The above means that for a sufficiently small gradient step, the gradient descent process is reversible. That is, we can always recover the previous model parameters given the current ones, assuming that the batch is fixed. We use the notion of reversibility throughout this paper. However, in practice we only have finite precision, thus instead of $\mathbb{R}$ we work with the finite set $\mathbb{F} \subset \mathbb{R}$. Furthermore, due to numerical stability issues, we do not have access to exact gradients, but only to approximate values $\widehat{\nabla f_A}$. For the rest of this paper, we assume these values are $L$-smooth on all elements in $\mathbb{F}^d$. That is,

$$\forall W_1, W_2 \in \mathbb{F}^d, A \subseteq X, \|\widehat{\nabla f_A}(W_1) - \widehat{\nabla f_A}(W_2)\| \leq L\|W_1 - W_2\|$$

This immediately implies that Lemma 2.1 holds even when precision is limited. Let us state the following theorem:

**Theorem 2.2.** *Let $W_1, W_2, ..., W_k \in \mathbb{F}^d \subset \mathbb{R}^d$, $A_1, A_2, ..., A_k \subseteq X$ and $\alpha < 1/L$. If it holds that $W_i = W_{i-1} - \alpha\widehat{\nabla f}_{A_{i-1}}(W_{i-1})$, then given $A_1, A_2, ..., A_{k-1}$ and $W_k$ we can retrieve $W_1$.*

*Proof.* Given $W_k$ we iterate over all $W \in \mathbb{F}^d$ until we find $W$ such that $W_k = W - \alpha\widehat{\nabla f}_{A_{i-1}}(W)$. Using Lemma 2.1, there is only a single element such that this equality holds, and thus $W = W_{k-1}$. We repeat this process until we retrieve $W_1$. $\square$

**SGD** We analyze the classic SGD algorithm presented in Algorithm 1. One difference to note in our algorithm, compared to the standard implementation, is the termination condition when the accuracy on the dataset is 100%. In practice the termination condition is not used, however, we only use it to prove that at *some point* in time the accuracy of the model is 100%.

---

**Algorithm 1:** SGD

1 $i \leftarrow 1$ // epoch counter
2 $W_{1,1}$ is an initial model
3 **while** *True* **do**
4      Take a random permutation of $X$, divided into batches $\{B_{i,j}\}_{j=1}^{n/b}$
5      **for** *j from 1 to n/b* **do**
6          **if** $acc(W_{i,j}, X) = 1$ **then** Return $W_{i,j}$
7          $W_{i,j+1} \leftarrow W_{i,j} - \alpha\nabla f_{B_{i,j}}(W_{i,j})$
8      $i \leftarrow i + 1, W_{i,1} \leftarrow W_{i-1,n/b+1}$

---

**Kolmogorov complexity** The Kolmogorov complexity of a string $x \in \{0, 1\}^*$, denoted by $K(x)$, is defined as the size of the smallest prefix Turing machine which outputs this string. We note that this definition depends on which encoding of Turing machines we use. However, one can show that this will only change the Kolmogorov complexity by a constant factor (Li and Vitányi, 2019).

We also use the notion of conditional Kolmogorov complexity, denoted by $K(x \mid y)$. This is the length of the shortest prefix Turing machine which gets $y$ as an auxiliary input and prints $x$. Note that the length of $y$ does not count towards the size of the machine which outputs $x$. So it can be the case that $|x| \ll |y|$ but it holds that $K(x \mid y) < K(x)$. We can also consider the Kolmogorov complexity of functions. Let $g : \{0, 1\}^* \rightarrow \{0, 1\}^*$ then $K(g)$ is the size of the smallest Turing machine which computes the function $g$.

The following properties of Kolmogorov complexity will be of use. Let $x, y, z$ be three strings:

- Extra information: $K(x \mid y, z) \leq K(x \mid z) + O(1) \leq K(x, y \mid z) + O(1)$
- Subadditivity: $K(xy \mid z) \leq K(x \mid z, y) + K(y \mid z) + O(1) \leq K(x \mid z) + K(y \mid z) + O(1)$

Random strings have the following useful property (Li and Vitányi, 2019):

**Theorem 2.3.** *For an $n$ bit string $x$ chosen uniformly at random, and some string $y$ independent of $x$ (i.e., $y$ is fixed before $x$ is chosen) and any $c \in \mathbb{N}$ it holds that $Pr[K(x \mid y) \geq n - c] \geq 1 - 1/2^c$.*

**Entropy and KL-divergence** Our proofs make extensive use of binary entropy and KL-divergence. In what follows we define these concepts and provide some useful properties.

*Entropy*: For $p \in [0, 1]$ we denote by $h(p) = -p \log p - (1 - p) \log(1 - p)$ the entropy of $p$. Note that $h(0) = h(1) = 0$.

*KL-divergence*: For $p, q \in (0, 1)$ let $D_{KL}(p \parallel q) = p \log \frac{p}{q} + (1 - p) \log \frac{1-p}{1-q}$ be the Kullback Leibler divergence (KL-divergence) between two Bernoulli distributions with parameters $p, q$. We also extend the above for the case where $q, p \in \{0, 1\}$ as follows: $D_{KL}(1 \parallel q) = D_{KL}(0 \parallel q) = 0, D_{KL}(p \parallel 1) = \log(1/p), D_{KL}(p \parallel 0) = \log(1/(1 - p))$. This is just notation that agrees with Lemma 2.4. We also state the following result of Pinsker's inequality applied to Bernoulli random variables: $D_{KL}(p \parallel q) \geq 2(p - q)^2$.

**Representing sets** Let us state some useful bounds on the Kolmogorov complexity of sets. A more detailed explanation regarding the Kolmogorov complexity of sets and permutations together with the proof to the lemma below appears in Appendix A.

**Lemma 2.4.** *Let $A \subseteq B, |B| = m, |A| = \gamma m$, and let $g : B \rightarrow \{0, 1\}$. For any set $Y \subseteq B$ let $Y_1 = \{x \mid x \in Y, g(x) = 1\}, Y_0 = Y \setminus Y_1$ and $\kappa_Y = \frac{|Y_1|}{|Y|}$. It holds that*

$$K(A \mid B, g) \leq m\gamma(\log(e/\gamma) - D_{KL}(\kappa_B \parallel \kappa_A)) + O(\log m)$$

## 3 ACCURACY DISCREPANCY

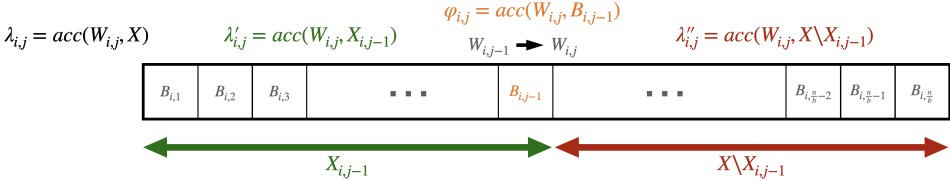

Figure 1: A visual summary of our notations.

First, let us define some useful notation ($W_{i,j}, B_{i,j}$ are formally defined in Algorithm 1):

- $\lambda_{i,j} = acc(W_{i,j}, X)$. This is the accuracy of the model in epoch $i$ on the entire dataset $X$, *before* performing the GD step on batch $j$.
- $\varphi_{i,j} = acc(W_{i,j}, B_{i,j-1})$. This is the accuracy of the model on the $(j - 1)$-th batch in the $i$-th epoch *after* performing the GD step on the batch.

- $X_{i,j} = \bigcup_{k=1}^{j} B_{i,k}$ (note that $\forall i, X_{i,0} = \emptyset, X_{i,n/b} = X$). This is the set of elements in the first $j$ batches of epoch $i$. Let us also denote $n_j = |X_{i,j}| = jb$ (Note that $\forall j, i_1, i_2, |X_{i_1,j}| = |X_{i_2,j}|$, thus $i$ need not appear in the subscript).

- $\lambda'_{i,j} = acc(W_{i,j}, X_{i,j-1}), \lambda''_{i,j} = acc(W_{i,j}, X \setminus X_{i,j-1})$, where $\lambda'_{i,j}$ is the accuracy of the model on the set of all previously seen batch elements, *after* performing the GD step on the $(j-1)$-th batch and $\lambda''_{i,j}$ is the accuracy of the same model, on all remaining elements ($j$-th batch onward). To avoid computing the accuracy on empty sets, $\lambda'_{i,j}$ is defined for $j \in [2, n/b + 1]$ and $\lambda''_{i,j}$ is defined for $j \in [1, n/b]$.

- $\rho_{i,j} = D_{KL}(\lambda'_{i,j} \parallel \varphi_{i,j})$ is the *accuracy discrepancy* for the $j$-th batch in iteration $i$ and $\rho_i = \sum_{j=2}^{n/b+1} \rho_{i,j}$ is the *accuracy discrepancy* at iteration $i$.

In our analysis, we consider $t$ epochs of the SGD algorithm. Our goal for this section is to derive a connection between $\sum_{i=1}^{t} \rho_i$ and $t$.

**Bounding $t$:** Our goal is to use the entropy compression argument to show that if $\sum_{i=1}^{t} \rho_i$ is sufficiently large we can bound $t$. Let us start by formally defining the random bits which the algorithm uses. Let $r_i$ be the string of random bits representing the random permutation of $X$ at epoch $i$. As we consider $t$ epochs, let $r = r_1 r_2 \ldots r_t$.

Note that the number of bits required to represent an arbitrary permutation of $[n]$ is given by:
$$\lceil \log(n!) \rceil = n \log n - n \log e + O(\log n) = n \log(n/e) + O(\log n),$$
where in the above we used Stirling's approximation. Thus, it holds that $|r| = t(n \log(n/e) + O(\log n))$ and according to Theorem 2.3, with probability at least $1 - 1/n^2$ it holds that $K(r) \geq tn \log(n/e) - O(\log n)$.

In the following lemma we show how to use the model at every iteration to efficiently reconstruct the batch at that iteration, where the efficiency of reconstruction is expressed via $\rho_i$.

**Lemma 3.1.** *It holds w.h.p that $\forall i \in [t]$ that:* $K(r_i \mid W_{i+1,1}, X) \leq n \log \frac{n}{e} - b\rho_i + \frac{n}{b} \cdot O(\log n)$

*Proof.* Recall that $B_{i,j}$ is the $j$-th batch in the $i$-th epoch, and let $P_{i,j}$ be a permutation of $B_{i,j}$ such that the order of the elements in $B_{i,j}$ under $P_{i,j}$ is the same as under $r_i$. Note that given $X$, if we know the partition into batches and all permutations, we can reconstruct $r_i$. According to Theorem 2.2, given $W_{i,j}$ and $B_{i,j-1}$ we can compute $W_{i,j-1}$. Let us denote by $Y$ the encoding of this procedure. To implement $Y$ we need to iterate over all possible vectors in $\mathbb{F}^d$ and over batch elements to compute the gradients. To express this program we require auxiliary variables of size at most $O(\log \min\{d, b\}) = O(\log n)$. Thus it holds that $K(Y) = O(\log n)$. Let us abbreviate $B_{i,1}, B_{i,2}, \ldots, B_{i,j}$ as $(B_{i,k})_{k=1}^{j}$. We write the following.

$K(r_i \mid X, W_{i+1,1}) \leq K(r_i, Y \mid X, W_{i+1,1}) + O(1) \leq K(r_i \mid X, W_{i+1,1}, Y) + K(Y \mid X, W_{i+1,1}) + O(1)$

$\leq O(\log n) + K((B_{i,k}, P_{i,k})_{k=1}^{n/b} \mid X, W_{i+1,1}, Y)$

$\leq O(\log n) + K((B_{i,k})_{k=1}^{n/b} \mid X, W_{i+1,1}, Y) + K((P_{i,k})_{k=1}^{n/b} \mid X, W_{i+1,1}, Y)$

$\leq O(\log n) + K((B_{i,k})_{k=1}^{n/b} \mid X, W_{i+1,1}, Y) + \sum_{j=1}^{n/b} K(P_{i,j})$

Let us bound $K((B_{i,k})_{k=1}^{n/b} \mid X, W_{i+1,1}, Y)$ by repeatedly using the subadditivity and extra information properties of Kolmogorov complexity.

$K((B_{i,k})_{k=1}^{n/b} \mid X, Y, W_{i+1,1}) \leq K(B_{i,n/b} \mid X, W_{i+1,1}) + K((B_{i,k})_{k=1}^{n/b-1} \mid X, Y, W_{i+1,1}, B_{i,n/b}) + O(1)$

$\leq K(B_{i,n/b} \mid X, W_{i+1,1}) + K((B_{i,k})_{k=1}^{n/b-1} \mid X, Y, W_{i,n/b}, B_{i,n/b}) + O(1)$

$\leq K(B_{i,n/b} \mid X, W_{i+1,1}) + K(B_{i,n/b-1} \mid X, W_{i,n/b}, B_{i,n/b})$

$+ K((B_{i,k})_{k=1}^{n/b-2} \mid X, Y, W_{i,n/b-1}, B_{i,n/b}, B_{i,n/b-1}) + O(1)$

$\leq \ldots \leq O(\frac{n}{b}) + \sum_{j=1}^{n/b} K(B_{i,j} \mid X, W_{i,j+1}, (B_{i,k})_{k=j+1}^{n/b}) \leq O(\frac{n}{b}) + \sum_{j=1}^{n/b} K(B_{i,j} \mid X_{i,j}, W_{i,j+1})$

where in the transitions we used the fact that given $W_{i,j}, B_{i,j-1}$ and $Y$ we can retrieve $W_{i,j-1}$. That is, we can always bound $K(... \mid Y, W_{i,j}, B_{i,j-1}, ...)$ by $K(... \mid Y, W_{i,j-1}, B_{i,j-1}, ...) + O(1)$.

To encode the order $P_{i,j}$ inside each batch, $b \log(b/e) + O(\log b)$ bits are sufficient. Finally we get that: $K(r_i \mid X, W_{i+1,1}) \le O(\frac{n}{b}) + \sum_{j=1}^{n/b}[K(B_{i,j} \mid X_{i,j}, W_{i,j+1}) + b\log(b/e) + O(\log b)]$.

Let us now bound $K(B_{i,j-1} \mid X_{i,j-1}, W_{i,j})$. Knowing $X_{i,j-1}$ we know that $B_{i,j-1} \subseteq X_{i,j-1}$. Thus we need to use $W_{i,j}$ to compress $B_{i,j-1}$. Applying Lemma 2.4 with parameters $A = B_{i,j-1}, B = X_{i,j-1}, \gamma = b/n_{j-1}, \kappa_A = \varphi_{i,j}, \kappa_B = \lambda'_{i,j}$ and $g(x) = acc(W_{i,j}, x)$. We get the following:

$$K(B_{i,j-1} \mid X_{i,j-1}, W_{i,j}) \le b(\log(\frac{e \cdot n_{j-1}}{b}) - \rho_{i,j}) + O(\log n_{j-1})$$

Adding $b\log(b/e) + O(\log b)$ to the above, we get the following bound on every element in the sum:

$$b(\log(\frac{e \cdot n_{j-1}}{b}) - \rho_{i,j}) + b\log(b/e) + O(\log b) + O(\log n_{j-1}) \le b\log n_{j-1} - b\rho_{i,j} + O(\log n_{j-1})$$

Note that the most important term in the sum is $-b\rho_{i,j}$. That is, the more the accuracy of $W_{i,j}$ on the batch, $B_{i,j-1}$, differs from the accuracy of $W_{i,j}$ on the set of elements containing the batch, $X_{i,j-1}$, we can represent the batch more efficiently. Let us now bound the sum: $\sum_{j=2}^{n/b+1}[b\log n_{j-1} - b\rho_{i,j} + O(\log n_{j-1})]$. Let us first bound the sum over $b\log n_{j-1}$:

$$\sum_{j=2}^{n/b+1} b\log n_{j-1} = \sum_{j=1}^{n/b} b\log jb = \sum_{j=1}^{n/b} b(\log b + \log j)$$

$$= n\log b + b\log(n/b)! = n\log b + n\log\frac{n}{b\cdot e} + O(\log n) = n\log\frac{n}{e} + O(\log n)$$

Finally, we can write that:

$$K(r_i \mid X, W_{i+1,1}) \le O(\frac{n}{b}) + \sum_{j=2}^{n/b+1}[b\log n_{j-1} - b\rho_{i,j} + O(\log n)] \le n\log\frac{n}{e} - b\rho_i + \frac{n}{b}\cdot O(\log n)$$

$\square$

Using the above we know that when the value $\rho_i$ is sufficiently high, the random permutation of the epoch can be compressed. We use the fact that random strings are incompressible to bound $\frac{1}{t}\sum_{i=1}^{t}\rho_i$.

**Theorem 3.2.** *If the algorithm does not terminate by the $t$-th iteration, then it holds w.h.p that $\forall t, \frac{1}{t}\sum_{i=1}^{t}\rho_i \le O(\frac{n\log n}{b^2})$.*

*Proof.* Using arguments similar to Lemma 3.1, we can show that $K(r, W_{1,1} \mid X) \le K(W_{t+1,1}) + O(t) + \sum_{k=1}^{t} K(r_k \mid X, W_{k+1,1})$ (formally proved in Lemma A.3). Combining this with Lemma 3.1, we get that $K(r, W_{1,1} \mid X) \le K(W_{t+1,1}) + t[n(\log(n/e) + \frac{n\cdot O(\log n)}{b} - b\rho_i + O(\log n)]$.

Our proof implies that we can reconstruct not only $r$, but also $W_{1,1}$ using $X, W_{t+1,1}$. Due to the incompressibility of random strings, we get that w.h.p $K(r, W_{1,1} \mid X) \ge d + tn\log(n/e) - O(\log n)$. Combining the lower and upper bound for $K(r, W_{1,1} \mid X)$ we can get the following inequality:

$$d + tn\log(n/e) - O(\log n) \le d + t[n(\log(n/e) + \frac{n\cdot O(\log n)}{b} + O(\log n)] - \sum_{i=1}^{t} b\rho_i \quad (1)$$

$$\implies \frac{1}{t}\sum_{i=1}^{t}\rho_i \le \underbrace{\frac{n\cdot O(\log n)}{b^2} + \frac{O(\log n)}{b}}_{\beta(n,b)} + \frac{O(\log n)}{bt} = O(\frac{n\log n}{b^2}) \quad \square$$

Let $\beta(n, b)$ be the exact value of the asymptotic expression in Inequality 1. Theorem 3.2 says that as long as SGD does not terminate the average accuracy discrepeancy cannot be too high. Using the contra-positive we get the following useful corollary (proof is deferred to Appendix A.3).

**Corollary 3.3.** *If $\forall k, \frac{1}{k}\sum_{i=1}^{k}\rho_i > \beta(n, b) + \gamma$, for $\gamma = \Omega(b^{-1}\log n)$, then w.h.p SGD terminates within $O(1)$ epochs.*

**The case for weak models**  Using the above we can also derive some interesting negative results when the model is not expressive enough to get perfect accuracy on the data. It must be the case that the average accuracy discrepancy tends below $\beta(n, b)$ over time. We verify this experimentally on the MNIST dataset (Appendix B), showing that the average accuracy indeed drops over time when the model is weak compared to the dataset. We also confirm that the dependence of the threshold in $b$ is indeed inversely quadratic.

## 4 THE ROLE OF RANDOMNESS IN GD INITIALIZATION

Our goal for this section is to show that when the amount of randomness in the perturbation is too small, for any model architecture which is differentiable and $L$-smooth there are inputs for which Algorithm 2 requires exponential time to terminate, even for extremely overparameterized models.

**Perturbation families**  Let us consider a family of $2^\ell$ functions indexed by length $\ell$ real valued vectors $\Psi_\ell = \{\psi_z\}_{z \in \mathbb{R}^\ell}$. Recall that throughout this paper we assume finite precision, thus every $z$ can be represented using $O(\ell)$ bits. We say that $\Psi_\ell$ is a reversible perturbation family if it holds that $\forall z \in \mathbb{R}^\ell, \psi_z$ is one-to-one. We often use the notation $\Psi_\ell(W)$, which means pick $z \in \mathbb{R}^\ell$ uniformly at random, and apply $\psi_z(W)$. We often refer to $\Psi_\ell$ as simply a perturbation.

We note that the above captures a wide range of natural perturbations. For example $\psi_z(W) = W + W_z$ where $W_z[i] = z[i \mod \ell]$. Clearly $\psi_z(W)$ is reversible.

**Gradient descent**  The GD algorithm we analyze is formally given in Algorithm 2.

---

**Algorithm 2:** GD$(W, Y, \delta)$ **Input**: initial model $W$, dataset $Y$, desired accuracy $\delta$

1  $i = 1, T = o(2^m) + poly(d)$
2  $W = \Psi_\ell(W)$
3  **while** $acc(W, Y) < \delta$ *and* $i < T$ **do**
4  $\quad\quad W \leftarrow W - \alpha \nabla f_Y(W)$
5  $\quad\quad i \leftarrow i + 1$
6  Return $W$

---

Let us denote by $m$ the number of elements in $Y$. We make the following 2 assumptions for the rest of this section: (1) $\ell = o(m)$. (2) There exists $T = o(2^m) + poly(d)$ and a perturbation family $\Psi_\ell$ such that for every input $W, Y$ within $T$ iterations GD terminates and returns a solution that has at least $\delta$ accuracy on $Y$ with constant probability. We show that the above two assumptions cannot hold together. That is, if the amount of randomness is sublinear in $m$, there must be instances with exponential running time, even when $d \gg m$.

To show the above, we define a variant of SGD, which uses GD as a sub procedure (Algorithm 3). Assume that our data set is a binary classification task (it is easy to generalize our results to any number of classes), and that elements in $X$ are assigned random labels. Furthermore, let us assume that $d = o(n)$, e.g., $d = n^{0.99}$. It holds that w.h.p we cannot train a model with $d$ parameters that achieves any accuracy better than $1/2 + o(1)$ on $X$ (Lemma A.4). Let us take $\epsilon$ to be a small constant. We show that if assumptions 1 and 2 hold, then Algorithm 3 must terminate and return a model with $1/2 + \Theta(1)$ accuracy on $X$, leading to a contradiction. Our analysis follows the same line as the previous section, and uses the same notation.

**Reversibility**  First, we must show that Algorithm 3 is still reversible. Note that we can take the same approach as before, where the only difference is that in order to get $W_{i,j}$ from $W_{i,j+1}$ we must now get all the intermediate values from the call to GD. As the GD steps are applied to the same batch, this amounts to applying Lemma 2.1 several times instead of once per iteration. More specifically, we must encode for every batch a number $T_{i,j} = o(2^b) + poly(d) = o(2^b) + poly(n)$ (recall that $d = o(n)$) and apply Lemma 2.1 $T_{i,j}$ times.

This results in $\psi_z(W_{i,j})$. If we know $z, \Psi_\ell$ then we can retrieve $\psi_z$ and efficiently retrieve $W_{i,j}$ using only $O(\log d) = O(\log n)$ additional bits (by iterating over all values in $\mathbb{F}^d$). Therefore, in every

---

**Algorithm 3:** SGD'

1   $i \leftarrow 1$ // epoch counter
2   $W_{1,1}$ is an initial model
3   **while** *True* **do**
4      Take a random permutation of $X$, divided into batches $\{B_{i,j}\}_{j=1}^{n/b}$
5      **for** *j from 1 to n/b* **do**
6         **if** $acc(W_{i,j}, X) \geq 1/2(1 - \epsilon)$ **then** Return $W_{i,j}$
7         $W_{i,j+1} \leftarrow GD(W_{i,j}, B_{i,j}, \frac{1}{2(1-2\epsilon)})$
8      $i \leftarrow i + 1, W_{i,1} \leftarrow W_{i-1,n/b+1}$

---

iteration we have the following additional terms: $\log T + O(\log n) + \ell = o(b) + O(\log n)$. Summing over $n/b$ iterations we get $o(n)$ per epoch. We state the following Lemma analogous to Lemma 3.1.

**Lemma 4.1.** *For Algorithm 3 it holds w.h.p that $\forall i \in [t]$ that: $K(r_i \mid W_{i+1,1}, X, \Psi_\ell) \leq n \log \frac{n}{e} - b\rho_i + \beta(n, b) + o(n)$.*

We show that under our assumptions, Algorithm 3 must terminate, leading to a contradiction.

**Lemma 4.2.** *Algorithm 3 with $b = \Omega(\log n)$ terminates within $O(T)$ iterations w.h.p.*

*Proof.* Our goal is to lower bound $\rho_i = \sum_{j=2}^{n/b+1} D_{KL}(\lambda'_{i,j} \parallel \varphi_{i,j})$. Let us first upper bound $\lambda'_{i,j}$. Using the fact that $\lambda'_{i,j} \leq \frac{n\lambda_{i,j}}{(j-1)b}$ (Lemma A.5) combined with the fact that $\lambda_{i,j} \leq 1/2(1 - \epsilon)$ as long as the algorithm does not terminate, we get that $\forall j \in [2, n/b + 1]$ it holds that $\lambda'_{i,j} \leq \frac{n}{2(1-\epsilon)(j-1)b}$. Using the above we conclude that as long as we do not terminate it must hold that $\lambda'_{i,j} \leq \frac{1}{2(1-\epsilon)^2}$ whenever $j \in I = [(1 - \epsilon)n/b + 1, n/b + 1]$. That is, $\lambda'_{i,j}$ must be close to $\lambda_{i,j}$ towards the end of the epoch, and therefore must be sufficiently small. Note that $|I| \geq \epsilon n/b$.

We know that as long as the algorithm does not terminate it holds that $\varphi_{i,j} > 1/2(1 - 2\epsilon)$ with some constant probability. Furthermore, this probability is taken over the randomness used in the call to GD (the randomness of the perturbation). This fact allows us to use Hoeffding-type bounds for the $\varphi_{i,j}$ variables. If $\varphi_{i,j} > 1/2(1 - 2\epsilon)$ we say that it is *good*. Therefore in expectation a constant fraction of $\varphi_{i,j}, j \in I$ are good. Applying a Hoeffding type bound we get that w.h.p a constant fraction of $\varphi_{i,j}, j \in I$ are good. Denote these good indices by $I_g \subseteq I$. We are now ready to bound $\rho_i$.

$$\rho_i = \sum_{j=2}^{n/b+1} D_{KL}(\lambda'_{i,j} \parallel \varphi_{i,j}) \geq \sum_{j \in I_g} D_{KL}(\lambda'_{i,j} \parallel \varphi_{i,j}) \geq \sum_{j \in I_g} D_{KL}(\frac{1}{2(1-\epsilon)^2} \parallel \frac{1}{2(1-2\epsilon)})$$

$$\geq \Theta(\frac{n}{b}) \cdot \epsilon(\frac{1}{2(1-2\epsilon)} - \frac{1}{2(1-\epsilon)^2})^2 = \Theta(\frac{n}{b}) \cdot \epsilon^5 = \Theta(\frac{n}{b})$$

Where in the transitions we used the fact that KL-divergence is non-negative, and Pinsker's inequality. Finally, requiring that $b = \Omega(\log n)$ we get that $b\rho_i - \beta(n, b) - o(n) = \Theta(n) - \Theta(\frac{n \log n}{\log^2 n}) - o(n) = \Theta(n)$. Following the same calculation as in Corollary 3.3, this guarantees termination within $O(\frac{\log n}{n})$ epochs, or $O(T \cdot \frac{n}{b} \cdot \frac{\log n}{n}) = O(T)$ iterations (gradient descent steps). $\qquad \square$

The above leads to a contradiction. It is critical to note that the above does not hold if $T = 2^m = 2^b$ or if $\ell = \Theta(n)$, as both would imply that the $o(n)$ term becomes $\Theta(n)$. We state our main theorem:

**Theorem 4.3.** *For any differentiable and L-smooth model class with $d$ parameters and a perturbation class $\Psi_\ell$ such that $\ell = o(m)$ there exist an input data set $Y$ of size $m$ such that GD requires $\Omega(2^m)$ iterations to achieve $\delta$ accuracy on $Y$, even if $\delta = 1/2 + \Theta(1)$ and $d \gg m$.*

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

## A  OMITTED PROOFS AND EXPLENATIONS

### A.1  REPRESENTING SETS AND PERMUTATIONS

Throughout this paper, we often consider the value $K(A)$ where $A$ is a set. Here the program computing $A$ need only output the elements of $A$ (in any order). When considering $K(A \mid B)$ such that $A \subseteq B$, it holds that $K(A \mid B) \leq \lceil \log \binom{|B|}{|A|} \rceil + O(\log |B|)$. To see why, consider Algorithm 4. In the algorithm $i_A$ is the index of $A$ when considering some ordering of all subsets of $B$ of size $|A|$. Thus $\lceil \log \binom{|B|}{|A|} \rceil$ bits are sufficient to represent $i_A$. The remaining variables $i, m_A, m_B$ and any

---

**Algorithm 4:** Compute $A$ given $B$ as input

1   $m_A \leftarrow |A|, m_B \leftarrow |B|, i \leftarrow 0, i_A$ is a target index
2   **for** *every subset $C \subseteq B$ s.t $|C| = m_A$ (in a predetermined order)* **do**
3      **if** $i = i_A$ **then** Print $C$
4      $i \leftarrow i + 1$

---

additional variables required to construct the set $C$ are all of size at most $O(\log |B|)$ and there is at most a constant number of them.

During our analysis, we often bound the Kolmogorov complexity of tuples of objects. For example, $K(A, P \mid B)$ where $A \subseteq B$ is a set and $P : A \rightarrow [|A|]$ is a permutation of $A$ (note that $A, P$ together form an ordered tuple of the elements of $A$). Instead of explicitly presenting a program such as Algorithm 4, we say that if $K(A \mid B) \leq c_1$ and $c_2$ bits are sufficient to represent $P$, thus $K(A, P \mid B) \leq c_1 + c_2 + O(1)$. This just means that we directly have a variable encoding $P$ into the program that computes $A$ given $B$ and uses it in the code. For example, we can add a permutation to Algorithm 4 and output an ordered tuple of elements rather than a set. Note that when representing a permutation of $A, |A| = k$, instead of using functions, we can just talk about values in $\lceil \log k! \rceil$. That is, we can decide on some predetermined ordering of all permutations of $k$ elements, and represent a permutation as its number in this ordering.

## A.2   OMITTED PROOFS FOR SECTION 2

**Lemma A.1.** *For $p \in [0, 1]$ it holds that $h(p) \leq p \log(e/p)$.*

*Proof.* Let us write our lemma as:

$$h(p) = -p \log p - (1 - p) \log(1 - p) \leq p \log(e/p)$$

Rearranging we get:

$$- (1 - p) \log(1 - p) \leq p \log p + p \log(1/p) + p \log e$$
$$\implies -(1 - p) \log(1 - p) \leq p \log e$$
$$\implies - \ln(1 - p) \leq \frac{p}{(1 - p)}$$

Note that $- \ln(1 - p) = \int_0^p \frac{1}{(1-x)} dx \leq p \cdot \frac{1}{(1-p)}$. Where in the final transition we use the fact that $\frac{1}{(1-x)}$ is monotonically increasing on $[0, 1]$. This completes the proof. $\square$

**Lemma A.2.** *For $p, \gamma, q \in [0, 1]$ where $p\gamma \leq q, (1 - p)\gamma \leq (1 - q)$ it holds that*

$$qh(\frac{p\gamma}{q}) + (1 - q)h(\frac{(1 - p)\gamma}{(1 - q)}) \leq h(\gamma) - \gamma D_{KL}(p \parallel q)$$

*Proof.* Let us expand the left hand side using the definition of entropy:

$$qh(\frac{p\gamma}{q}) + (1-q)h(\frac{(1-p)\gamma}{(1-q)})$$

$$= -q(\frac{p\gamma}{q}\log\frac{p\gamma}{q} + (1-\frac{p\gamma}{q})\log(1-\frac{p\gamma}{q}))$$

$$- (1-q)(\frac{(1-p)\gamma}{(1-q)}\log\frac{(1-p)\gamma}{(1-q)} + (1-\frac{(1-p)\gamma}{(1-q)})\log(1-\frac{(1-p)\gamma}{(1-q)}))$$

$$= -(p\gamma\log\frac{p\gamma}{q} + (q-p\gamma)\log\frac{q-p\gamma}{q})$$

$$- ((1-p)\gamma\log\frac{(1-p)\gamma}{(1-q)} + ((1-q)-(1-p)\gamma)\log\frac{(1-q)-(1-p)\gamma}{1-q})$$

$$= -\gamma\log\gamma - \gamma D_{KL}(p \parallel q)$$

$$- (q-p\gamma)(\log\frac{q-p\gamma}{q}) - ((1-q)-(1-p)\gamma)\log\frac{(1-q)-(1-p)\gamma}{1-q})$$

Where in the last equality we simply sum the first terms on both lines. To complete the proof we use the log-sum inequality for the last expression. The log-sum inequality states that: Let $\{a_k\}_{k=1}^m, \{b_k\}_{k=1}^m$ be *non-negative* numbers and let $a = \sum_{k=1}^m a_k, b = \sum_{k=1}^m b_k$, then $\sum_{k=1}^m a_i\log\frac{a_i}{b_i} \geq a\log\frac{a}{b}$. We apply the log-sum inequality with $m = 2, a_1 = q - p\gamma, a_2 = (1-q) - (1-p)\gamma, a = 1 - \gamma$ and $b_1 = q, b_2 = 1 - q, b = 1$, getting that:

$$(q-p\gamma)(\log\frac{q-p\gamma}{q}) + ((1-q)-(1-p)\gamma)\log\frac{(1-q)-(1-p)\gamma}{1-q}) \geq (1-\gamma)\log(1-\gamma)$$

Putting everything together we get that

$$-\gamma\log\gamma - \gamma D_{KL}(p \parallel q)$$

$$- (q-p\gamma)(\log\frac{q-p\gamma}{q}) - ((1-q)-(1-p)\gamma)\log\frac{(1-q)-(1-p)\gamma}{1-q})$$

$$\leq -\gamma\log\gamma - (1-\gamma)\log(1-\gamma) - \gamma D_{KL}(p \parallel q) = h(\gamma) - \gamma D_{KL}(p \parallel q)$$

$\square$

**Lemma 2.4.** *Let* $A \subseteq B, |B| = m, |A| = \gamma m$, *and let* $g : B \to \{0,1\}$. *For any set* $Y \subseteq B$ *let* $Y_1 = \{x \mid x \in Y, g(x) = 1\}, Y_0 = Y \setminus Y_1$ *and* $\kappa_Y = \frac{|Y_1|}{|Y|}$. *It holds that*

$$K(A \mid B, g) \leq m\gamma(\log(e/\gamma) - D_{KL}(\kappa_B \parallel \kappa_A)) + O(\log m)$$

*Proof.* The algorithm is very similar to Algorithm 4, the main difference is that we must first compute $B_1, B_0$ from $B$ using $g$, and select $A_1, A_0$ from $B_1, B_0$, respectively, using two indices $i_{A_1}, i_{A_0}$. Finally we print $A = A_1 \cup A_0$. We can now bound the number of bits required to represent $i_{A_1}, i_{A_0}$. Note that $|B_1| = \kappa_B m, |B_0| = (1-\kappa_B)m$. Note that for $A_1$ we pick $\gamma\kappa_A m$ elements from $\kappa_B m$ elements and for $A_0$ we pick $\gamma(1-\kappa_A)m$ elements from $(1-\kappa_B)m$ elements. The number of bits required to represent this selection is:

$$\lceil\log\binom{\kappa_B m}{\gamma\kappa_A m}\rceil + \lceil\log\binom{(1-\kappa_B)m}{\gamma(1-\kappa_A)m}\rceil \leq \kappa_B mh(\frac{\gamma\kappa_A}{\kappa_B}) + (1-\kappa_B)mh(\frac{\gamma(1-\kappa_A)}{(1-\kappa_B)})$$

$$\leq m(h(\gamma) - \gamma D_{KL}(\kappa_B \parallel \kappa_A)) \leq m\gamma(\log(e/\gamma) - D_{KL}(\kappa_B \parallel \kappa_A))$$

Where in the first inequality we used the fact that $\forall 0 \leq k \leq n, \log\binom{n}{k} \leq nh(k/n)$, Lemma A.2 in the second transition, and Lemma A.1 in the third transition. Note that when $\kappa_A = 0, 1$ We only have one term of the initial sum. For example, for $\kappa_A = 1$ we get:

$$\lceil\log\binom{\kappa_B m}{\gamma\kappa_A m}\rceil = \lceil\log\binom{\kappa_B m}{\gamma m}\rceil \leq \kappa_B mh(\frac{\gamma}{\kappa_B})$$

$$\leq m\gamma\log(e\kappa_B/\gamma) = m\gamma(\log(e/\gamma) - \log(1/\kappa_B))$$

And similar computation yields $m\gamma(\log(e/\gamma) - \log(1/(1-\kappa_B)))$ for $\kappa_A = 0$. Finally, the additional $O(\log m)$ factor is due to various counters and variables, similarly to Algorithm 4. $\square$

### A.3 OMITTED PROOFS FOR SECTION 3

**Lemma A.3.** *It holds that $K(r, W_{1,1} \mid X) \leq K(W_{t+1,1}) + O(t) + \sum_{k=1}^{t} K(r_k \mid X, W_{k+1,1})$.*

*Proof.* Similarly to the definition of $Y$ in Lemma 3.1, let $Y'$ be the program which receives $X, r_i, W_{i+1,1}$ as input and repeatedly applies Theorem 2.2 to retrieve $W_{i,1}$. As $Y'$ just needs to reconstruct all batches from $X, r_i$ and call $Y$ for $n/b$ times, it holds that $K(Y') = O(\log n)$. Using the subadditivity and extra information properties of $K()$, together with the fact that $W_{1,1}$ can be reconstructed given $X, W_{t+1,1}, Y'$, we write the following:

$$K(r \mid X) \leq K(r, W_{1,1}, Y', W_{t+1,1} \mid X) + O(1) \leq K(W_{1,1,}, W_{t+1,1}, Y' \mid X) + K(r \mid X, Y', W_{t+1,1}) + O(1)$$
$$\leq K(W_{t+1,1} \mid X) + K(r \mid X, Y', W_{t+1,1}) + O(\log n)$$

First, we note that: $\forall i \in [t-1], K(r_i \mid X, Y', W_{i+2,1}, r_{i+1}) \leq K(r_i \mid X, Y', W_{i+1,1}) + O(1)$. Where in the last inequality we simply execute $Y'$ on $X, W_{i+2,1}, r_{i+1}$ to get $W_{i+1,1}$. Let us write:

$$K(r_1 r_2 \ldots r_t \mid X, Y', W_{t+1,1}) \leq K(r_t \mid X, Y', W_{t+1,1}) + K(r_1 r_2 \ldots r_{t-1} \mid X, Y', W_{t+1,1}, r_t) + O(1)$$
$$\leq K(r_t \mid X, W_{t+1,1}) + K(r_1 r_2 \ldots r_{t-1} \mid X, Y', W_{t,1}) + O(1)$$
$$\leq K(r_t \mid X, W_{t+1,1}) + K(r_{t-1} \mid X, W_{t,1}) + K(r_1 r_2 \ldots r_{t-2} \mid X, Y', W_{t-1,1}) + O(1)$$

$$\leq \cdots \leq O(t) + \sum_{k=1}^{t} K(r_k \mid X, W_{k+1,1})$$

Combining everything together we get that:

$$K(r \mid X) \leq K(W_{t+1,1}) + O(t) + \sum_{k=1}^{t} K(r_k \mid X, W_{k+1,1})$$

$\square$

**Corollary 3.3.** *If $\forall k, \frac{1}{k} \sum_{i=1}^{k} \rho_i > \beta(n, b) + \gamma$, for $\gamma = \Omega(b^{-1} \log n)$, then w.h.p SGD terminates within $O(1)$ epochs.*

*Proof.* Let us simplify Inequality 1.

$$d + tn \log(n/e) - O(\log n) \leq d + t[n(\log(n/e) + \frac{n \cdot O(\log n)}{b} + O(\log n)] - \sum_{i=1}^{t} b\rho_i$$

$$\implies -O(\log n) \leq t[\frac{n \cdot O(\log n)}{b} + O(\log n)] - \sum_{i=1}^{t} b\rho_i$$

$$\implies (\sum_{i=1}^{t} \rho_i) - t\beta(n, b) \leq O(\log n)/b$$

Our condition implies that $\sum_{i=1}^{t} \rho_i > t(\beta(n, b) + \gamma)$. This allows us to rewrite the above inequality as:

$$t\gamma \leq O(\log n)/b \implies t = O(1)$$

$\square$

### A.4 OMITTED PROOFS FOR SECTION 4

**Lemma A.4.** *Let $X$ be some set of size $n$ and let $f : X \to \{0, 1\}$ be a random binary function. It holds w.h.p that there exists no function $g : X \to \{0, 1\}$ such that $K(g \mid X) = o(n)$ and $g$ agrees with $f$ on $n(1/2 + \Theta(1))$ elements in $X$.*

*Proof.* Let us assume that $g$ agrees with $f$ on all except $\epsilon n$ elements in $X$ and bound $\epsilon$. Using Theorem 2.3, it holds w.h.p that $K(f \mid X) > n - O(\log n)$. We show that if $\epsilon$ is sufficiently far from $1/2$, we can use $g$ to compress $f$ below its Kolmogorov complexity, arriving at a contradiction.

We can construct $f$ using $g$ and the set of values on which they do not agree, which we denote by $D$. This set is of size $\epsilon n$ and therefore can be encoded using $\log \binom{n}{\epsilon n} \le nh(\epsilon)$ bits (recall that $\forall 0 \le k \le n, \log \binom{n}{k} \le nh(k/n)$) given $X$ (i.e., $K(D \mid X) \le nh(\epsilon)$). To compute $f(x)$ using $D, g$ we simply check if $x \in D$ and output $g(x)$ or $1 - g(x)$ accordingly. The total number of bits required for the above is $K(g, D \mid X) \le o(n) + nh(\epsilon)$ (where auxiliary variables are subsumed in the $o(n)$ term). We conclude that $K(f \mid X) \le o(n) + nh(\epsilon)$. Combining the upper and lower bounds on $K(f \mid X)$, it must hold that $o(n) + nh(\epsilon) \ge n - O(\log n) \implies h(\epsilon) \ge 1 - o(1)$. This inequality only holds when $\epsilon = 1/2 + o(1)$.

$\square$

**Lemma A.5.** *It holds that* $1 - \frac{n(1-\lambda_{i,j})}{(j-1)b} \le \lambda'_{i,j} \le \frac{n\lambda_{i,j}}{(j-1)b}$.

*Proof.* We can write the following for $j \in [2, n/b + 1]$:

$$n\lambda_{i,j} = \sum_{x \in X} acc(W_{i,j}, x) = \sum_{x \in X_{i,j-1}} acc(W_{i,j}, x) + \sum_{x \in X \setminus X_{i,j-1}} acc(W_{i,j}, x)$$

$$= (j-1)b\lambda'_{i,j} + (n - (j-1)b)\lambda''_{i,j}$$

$$\implies \lambda'_{i,j} = \frac{n\lambda_{i,j} - (n - (j-1)b)\lambda''_{i,j}}{(j-1)b}$$

Setting $\lambda''_{i,j} = 0$ we get

$$\lambda'_{i,j} = \frac{n\lambda_{i,j} - (n - (j-1)b)\lambda''_{i,j}}{(j-1)b} \le \frac{n\lambda_{i,j}}{(j-1)b}$$

And setting $\lambda''_{i,j} = 1$ we get

$$\lambda'_{i,j} = \frac{n\lambda_{i,j} - (n - (j-1)b)\lambda''_{i,j}}{(j-1)b} \ge 1 - \frac{n(1-\lambda_{i,j})}{(j-1)b}$$

$\square$

# B  EXPERIMENTS

**Experimental setup**    We perform experiments on MNIST dataset and the same data set with random labels (MNIST-RAND). We use SGD with learning rate 0.01 without momentum or regularization. We use a simple fully connected architecture with a single hidden layer, GELU activation units (a differentiable alternative to ReLU) and cross entropy loss. We run experiments with a hidden layer of size $2, 5, 10$. We consider batches of size $50, 100, 200$. For each of the datasets we run experiments for all configurations of architecture sizes and batch sizes for 300 epochs.

**Results**    Figure 2 and Figure 3 show the accuracy discrepancy and accuracy over epochs for all configurations for MNIST and MNIST-RAND respectively. Figure 4 and Figure 5 show for every batch size the accuracy discrepancy of all three model sizes on the same plot. All of the values displayed are averaged over epochs, i.e., the value for epoch $t$ is $\frac{1}{t}\sum_i x_i$.

First, we indeed observe that the scale of the accuracy discrepancy is inversely quadratic in the batch size, as our analysis suggests. Second, for MNIST-RAND we can clearly see that the average accuracy discrepancy tends below a certain threshold over time, where the threshold appears to be independent of the number of model parameters. We see similar results for MNIST when the model is small, but not when it is large. This is because the model does not reach its capacity within the timeframe of our experiment.

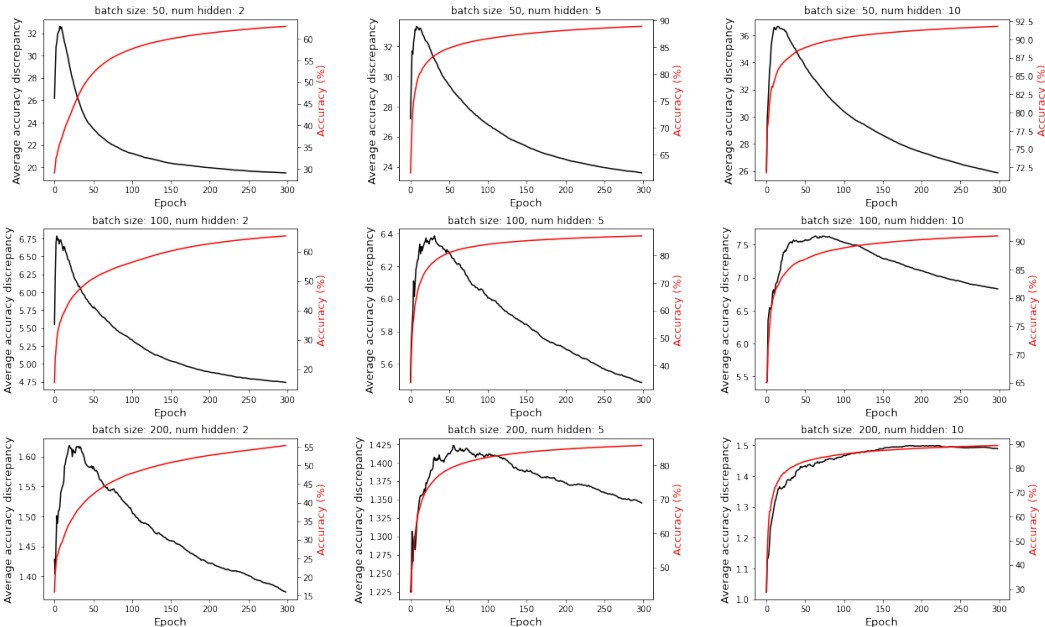

Figure 2: Full results for the MNIST dataset.

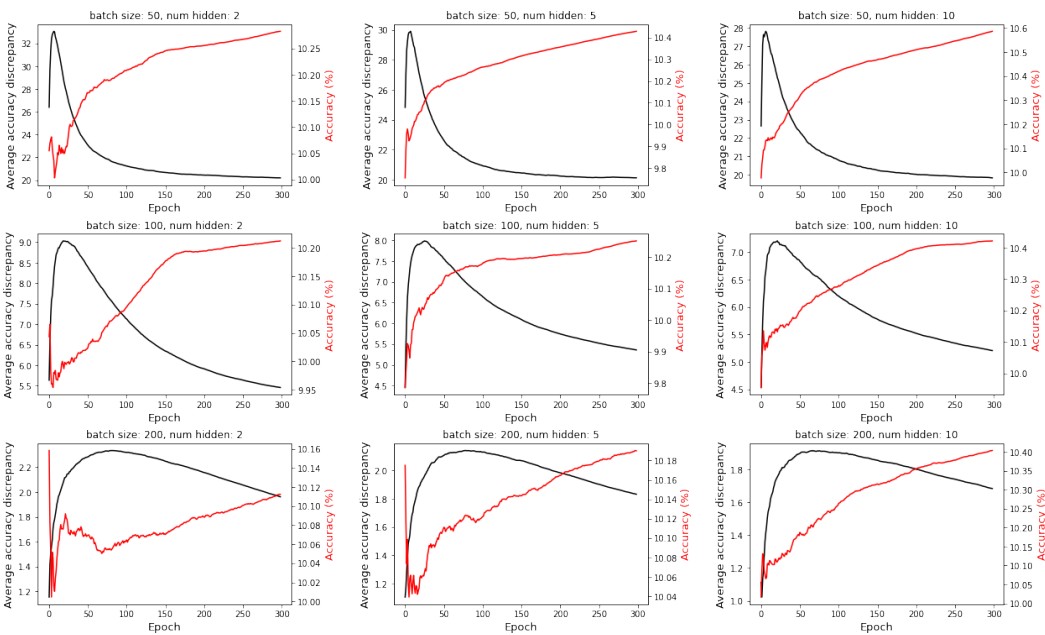

Figure 3: Full results for the MNIST-RAND dataset.

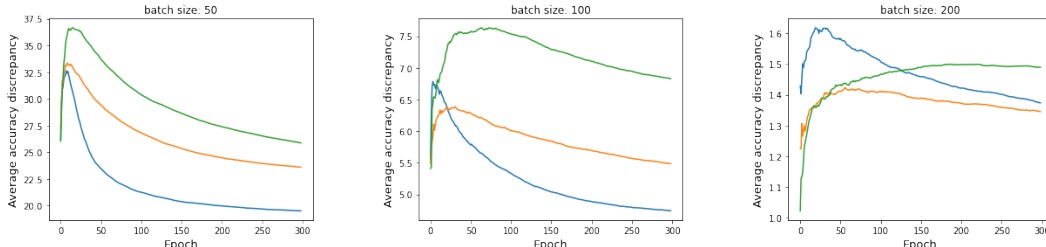

Figure 4: We plot for every batch size the accuracy discrepancies of the 3 different models. Results for MNIST.

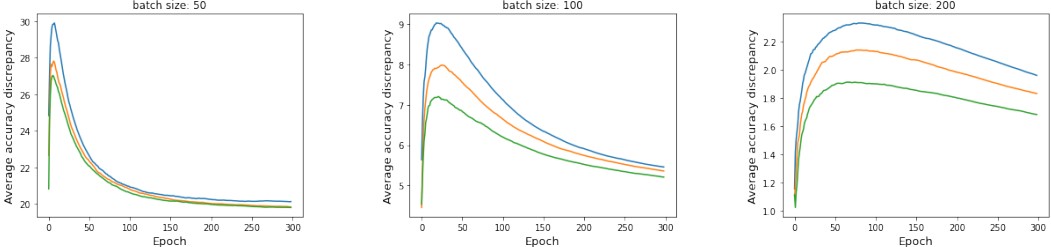

Figure 5: We plot for every batch size the accuracy discrepancies of the 3 different models. Results for MNIST-RAND.

