# OpenReview forum: "SGD Through the Lens of Kolmogorov Complexity"
_ICLR.cc/2023/Conference — Submitted to ICLR 2023_

### Official Review · Reviewer_T4KT · 2022-10-14

**Confidence:** 3
**Correctness:** 3
**Technical Novelty And Significance:** 3
**Empirical Novelty And Significance:** Not applicable
**Recommendation:** 5

**Clarity, Quality, Novelty And Reproducibility:**

* **Clarity**: As mentioned in the weaknesses above, the paper is not very clear and is hard to read. \
However, there is time for a careful revision of the manuscript to improve these issues.

* **Quality**: The quality of the paper is good. The ideas are interesting and well-executed.

* **Novelty**: As far as I know, employing Kolmogorov complexity (or similar compression arguments) to show the convergence of SGD or other algorithms is novel. This could spark new analyses of different algorithms.

* **Reproducibility**: \
*Theory*: I reproduced all the proofs except for the questions that I placed the authors in the weaknesses. I believe that after they answer them and clarify those in the manuscript every theoretical aspect should be reproducible. \
Some further notes in this regard are:
  * In the paragraph *Bounding $t$*: is not the probability $1 - 1/n^t$. Of course, this also implies a probability of at least $1 - 1/n^2$, but I believe that that is what one obtains after applying Theorem 2.3.
  * I could not find an open-access resource for the Kolmogorov complexity property $K(x|z) + \mathcal{O}(1) \leq K(x,y|z) + \mathcal{O}(1)$ for any three strings $x,y,z$. Therefore, I trusted this inequality as an axiom to reproduce their results. Could the authors link to an open-access source for this result?

  *Experiments*: There is no code to reproduce the experiments. They are easy to replicate due to the small size of the datasets and the simplicity of the experiment. Nonetheless, it would be helpful to include this resource.

**Strength And Weaknesses:**

**Strengths**

* The results in this paper are architecture-independent. \
This is good since the current understanding of the global convergence of SGD is for specific models that usually require over-parameterization. The only requirements for this theory are that the loss is differentiable, $L$-smooth, and that the step size is smaller than $1/L$, all of which are reasonable.

* The result of the necessity of superlinear randomness with respect to the number of samples for the convergence of GD is insightful. It opens a bridge between why SGD generalizes and GD does not in certain situations and opens the door to the design of the randomness in other versions of GD where the randomization is included differently such as stochastic gradient Langevin dynamics, or even the design of new GD-based stochastic algorithms. \
I believe that it could be beneficial for the article to stress this fact since the importance of this result seems overshadowed by Section 3.

**Weaknesses**

* The main statement of the convergence of SGD says that if the average accuracy discrepancies until epoch $t$ is larger than a certain threshold, then the algorithm converges within $\mathcal{O}(1)$ epochs. \
This statement is interesting on its own in the sense that it gives a sufficient metric to determine the convergence of SGD. That is, if one can see when the average accuracy discrepancy will exceed the threshold, then one knows when SGD will converge. \
However, it still does not give any convergence result for SGD, since it is still needed to run the algorithm to be able to calculate the accuracy discrepancies. That is, to determine the quantity that tells if SGD converges, one needs to run SGD until convergence.

  * The paper may be improved with an acknowledgement of this fact and placing the contribution in a different fashion: that is, as a finding of a sufficient condition for SGD convergence. \
  Otherwise, and more strongly, the paper would benefit from an analysis of the accuracy discrepancy and when it exceeds said threshold. Though I understand this may be the work of a future follow-up project.

* The paper is hard to read and sometimes leaves some useful information unwritten.

  * In the related work section it mentions the work from *Zhang et al. (2022)* requires *certain assumptions*. It would be helpful to state such conditions. Similarly, when discussing *Jin et al. (2022)* it is said that the randomness required for the perturbations on this work is beyond the considered threshold. This claim is not recovered in Section 4, so we are to believe such a statement.

  * Theorem 2.3 deals with the conditional Kolmogorov complexity $K(x|y)$ before it is introduced int eh following paragraphs. It would be useful to describe this property of random strings right after the general properties of the Kolmogorov complexity.

  * How does one go from the first to the second line of equations in the proof of Lemma 3.1?

  * Also in the proof of Lema 3.1, the papepr goes back and forth with the notation $B_{i,j}$ knowing $X_{i,j}$ and $B_{i,j-1}$ knowing $X_{i,j-1}$. Although equivalent, it would aid readability to maintain one indexing. The same happens in the definition of the variables of interest in the beginning of Section 3.

  * In proof of the Corollary 3.3, what is the exact value of $\beta(n,b)$? Also, what is $\gamma$? I could not follow the proof without that knowledge.

  * In the proof of Lemma 4.2, how does one go from the first to the second line of equations? I understand that Pinsker's inequality is employed. My question regards the $\Theta(n/b)$ term, based on the equation it seems it says that $|I_g| = \epsilon \Theta(n/b)$. If so, how do we know that?

* The KL divergence (or relative entropy) $D_{\text{KL}}(p \lVert q)$ extension is unjustified. The extremes with respect to $p$ are set to zero when there are real finite limiting points, and the extremes with respect to $q$ are set to some finite value when they diverge.

  * A suggestion is to use the standard agreement that $0 \log 0 = 0$ by continuity and let $D_{\text{KL}}(1 \lVert q) = \log(1/q)$ and $D_{\text{KL}}(0 \lVert q) = \log(1/(1-q))$, and $D_{\text{KL}}(p \lVert 0) \to \infty$ and $D_{\text{KL}}(p \lVert 1) \to \infty$. This should not be a problem for your theory since the relative entropy is only employed in the accuracy disagreement and we have that if $\lambda_{i,j}' \neq 0,1$ then $\varphi_{i,j} \neq 0,1$

* In the Appendix B, it would be helpful if some of the figures had the y-axis adjusted to have a better visual comparison between them. Without a closer look, in Figure 2 for batch size 200 and 10 hidden units, one may get the opposite idea.

**Summary Of The Paper:**

This paper analyzes the convergence of stochastic gradient descent (SGD). It also concerns the study of the minimum amount of randomness that needs to be added to non-stochastic gradient descent (GD) so that it can converge.

First, it defines the *accuracy discrepancy* as the cumulative difference (measured with the relative entropy)  between the accuracy of the model in the batches seen until iteration $j$ and the accuracy of the model at iteration $j$ of a certain epoch.

Then, it shows that if the average of accuracy discrepancies over the last $t$ epochs is over a certain threshold, then SGD converges to 100\% accuracy with high probability (w.h.p.) within $\mathcal{O}(1)$ epochs. This result is architecture-independent, in contrast with the current literature.

* An extra contribution of this result is its analysis, since it is uncommon in this area. It relies on Kolmogorov complexity, which is a measure of compression. The idea is to consider the randomness used to select the permutation of the batches in SGD. Pure randomness cannot be compressed, thus giving a lower bound on the Kolmogorov complexity. After observing the model at a certain iteration and the data, such randomness can be compressed and the paper shows an upper bound that depends on the accuracy discrepancy. Combining those they show that w.h.p. if at epoch $t$ the algorithm did not terminate then the accuracy discrepancy is smaller than a certain threshold. Hence, if the accuracy discrepancy is larger than such a threshold, it must terminate.

Finally, they argue that GD does not converge if the amount of randomness included can be represented with $o(n)$ bits and the amount of epochs is $\text{poly}(d,n)$, where $d$ is the parameters' dimension and $n$ is the number of data samples. That is, if the randomness is sublinear with $n$, there is a dataset for which the amount of epochs needed for convergence is superpolynomial (e.g. exponential).

**Summary Of The Review:**

This studies and gives insights in two important phenomena in the convergence of gradient-based algorithms:

* If finds a sufficient condition for the convergence of SGD. As long as the accuracy discrepancy (introduced in this paper) is larger than a certain threshold the algorithm terminates within $\mathcal{O}(1)$ epochs. Hence, understanding when such a quantity crosses the threshold is enough to understand when SGD will converge. This can serve to derive new theory to characterize the convergence of SGD.
* It shows that a necessary condition to ensure GD convergence is to have some randomness with a superlinear Kolmogorov complexity with respect to the number of samples. This is an insighful result that can serve to better understand and design current algorithms and to develop new ones.

To obtain these results the paper employs Kolmogorov complexity, which is a novel technique in convergence analysis as far as I know.

The paper is not very clear and has some claims that need further support, as I mention in the weaknesses section of the review.

Due to the lack of clarity, I must lean to rejection at the moment. The reason is that unless some of my questions and concerns are addressed, I cannot verify that the proofs are correct. Nonetheless, I am open to changing my rating to accept once they are successfully answered. This is because even though the convergence analysis is not direct, I believe that the paper contributes to a better understanding of the convergence of gradient-based methods and introduces an interesting proof technique that can be used in further work.

**Minor comments and nitpicks that did not impact the score of the review**

* Usually, with high probability (w.h.p.) always means that the probability tends to 1. I am not so sure the footnote is needed, then. Also, in your case, you prove a stronger convergence rate than $1 - 1/n$.
* 7th line, page 2: (and wide) ~~the~~ GD.
* Shouldn't it be $acc(W,A): \mathbb{R}^d \times 2^X \to[0,1]$?
* Before Lemma 4.1, in the parenthesis shouldn't it be "(by iterating over all values of $\mathbb{F}^d$)"?
* In the proof of Lemma 4.2, it should be $D_{\text{KL}}(\lambda_{i,j}' \lVert \varphi_{i,j})$.
* Figures 4 and 5 are hard to see for a colorblind person. Could you change the colors, please?

---

> ### Author Response · Authors · 2022-11-09
> **Authors' response to Reviewer T4KT (1/2)**
>
> Thank you for taking the time to review this paper. We found your comments extremely valuable, both regarding the presentation and the technical side of the paper. We tried to follow all of your comments to improve the writing of the paper.
>
> Please find our detailed comments below (minor comments not addressed directly are fixed in the new revision).
>
> >The result of the necessity of superlinear randomness with respect to the number of samples for the convergence of GD is insightful. It opens a bridge between why SGD generalizes and GD does not in certain situations and opens the door to the design of the randomness in other versions of GD where the randomization is included differently such as stochastic gradient Langevin dynamics, or even the design of new GD-based stochastic algorithms.
> I believe that it could be beneficial for the article to stress this fact since the importance of this result seems overshadowed by Section 3.
>
> Thank you for this insight. We tried incorporating this into the introduction (paragraph before "outline of our techniques").
>
> >The paper may be improved with an acknowledgement of this fact and placing the contribution in a different fashion: that is, as a finding of a sufficient condition for SGD convergence.
> Otherwise, and more strongly, the paper would benefit from an analysis of the accuracy discrepancy and when it exceeds said threshold. Though I understand this may be the work of a future follow-up project.
>
> We added the fact that we provide a sufficient condition for SGD convergence in the intro. Indeed the next natural step would be to use this new measure to show convergence for specific model classes. We hope to explore this in future work.
>
> > In the related work section it mentions the work from Zhang et al. (2022) requires certain assumptions. It would be helpful to state such conditions. Similarly, when discussing Jin et al. (2022) it is said that the randomness required for the perturbations on this work is beyond the considered threshold. This claim is not recovered in Section 4, so we are to believe such a statement.
>
> We've tried clarifying both of these issues in the revision. Please let us know if it is still unclear.
>
> > Theorem 2.3 deals with the conditional Kolmogorov complexity  before it is introduced int eh following paragraphs. It would be useful to describe this property of random strings right after the general properties of the Kolmogorov complexity.
>
> We've moved this according to your suggetion.
>
> > How does one go from the first to the second line of equations in the proof of Lemma 3.1?
>
> We've added some extra lines of math in the proof. Hopefully, it is clear now.
>
> >In proof of the Corollary 3.3, what is the exact value of $\beta(n,b)$? Also, what is $\gamma$? I could not follow the proof without that knowledge.
>
> Indeed the proof was vague. We kept the asymptotic notation for brevity but highlighted the value of $\beta(n,b)$. We also added more details to the proof (now deferred to the appendix).
>
> >In the proof of Lemma 4.2, how does one go from the first to the second line of equations? I understand that Pinsker's inequality is employed. My question regards the  term, based on the equation it seems it says that $|I_g| = \epsilon \Theta(n/b)$. If so, how do we know that?
>
> Note that $I_g \subseteq I$ where $|I| = \epsilon n/b$. So in expectation we get that $|I_g| = \epsilon  \Theta(n/b)$. As every iteration in $I$ is good with at least some constant probability, regardless of previous iterations, and $I$ is sufficiently large we also get concentration around the mean. We've explicitly stated the size of $I$ in the text in the new revision.
>
> > The KL divergence (or relative entropy)  extension is unjustified.
>
> We believe that extension is required when $\lambda'_{i,j} \neq 0$ but $\varphi_i,j =0,1$, which may happen. The values of the extension result from the calculation in Lemma 2.4. The goal of this extension is merely to keep notation simple (instead of using the true KL-divergence, and breaking things into cases). Please let us know if you still think this is unnecessary, as we would also be happy to simply use the standard definition of KL-divergence.

---

> ### Author Response · Authors · 2022-11-09
> **Authors' response to Reviewer T4KT (2/2)**
>
> > In the paragraph Bounding $t$: is not the probability $1 - 1/n^t$.
>
> We believe the current statement is correct (by plugging in $c=2 \log n$  in Theorem 2.3). Not that it is applied only once for $r$, and not every $r_i$ separately.
>
> >I could not find an open-access resource for the Kolmogorov complexity property $K(x|z) + \mathcal{O}(1) \leq K(x,y|z) + \mathcal{O}(1)$ for any three strings. Therefore, I trusted this inequality as an axiom to reproduce their results. Could the authors link to an open-access source for this result?
>
> Please take a look here: http://www.scholarpedia.org/article/Algorithmic_complexity#Properties_of_prefix_complexity .
> The relevant section is "Properties of prefix complexity", property (5) "extra information". Right below it is stated that "and all (in)equalities hold within an additive constant." and "All (in)equalities remain valid if K is (further) conditioned under some z".
>
> > Usually, with high probability (w.h.p.) always means that the probability tends to 1. I am not so sure the footnote is needed, then. Also, in your case, you prove a stronger convergence rate than $1-1/n$.
>
> Several definitions are used in the literature (for example, reviewer dQLw uses a different definition). Our results indeed hold with a probably $1-1/n^c$ for an arbitrarily large constant $c$. We wrote it as we did to avoid introducing further notation.
>
> ## comments we did not address in the current revision
> - Indexing in Lemma 3.1 - We would like to leave it as is if possible. We are worried about introducing bugs in the proof due to indexing.
>
> - Code and figures - We would be happy to release the code and update the figures according to your suggestions, however, both might be somewhat time-consuming, so we will do it in the final version if it is OK with you.

---

> > ### Comment · Reviewer_T4KT · 2022-11-14
> > **Follow-up on the answers from the authors**
> >
> > Thank you for your response and adjustments based on my comments. I am happy with most of them. Let me follow up on those for which I still have some remarks:
> >
> > * *Regarding the code and figures:* I understand it may require further work, and it seems okay to me to wait for the axis-aligned figures until the final version. Nonetheless, I expect some code (even if only preliminary) before that.
> >
> > * *Regarding the specifying of Zhang et al. (2020)*: It seems okay to me, but maybe re-write it as follows: "(e.g., considering a fully connected architecture with sub-differentiable and coordinate-wise Lipschitz activations and weights laying on a compact set)".
> >
> > * *Regarding Jin et al. (2022)*: It improved but is still unclear to me. It mentions that each perturbation **may** require a linear amount of randomness. Is it possible to show that the overall randomness **must** be at least linear? You may add a note in the appendix for this purpose if necessary.
> >
> > * *Regarding the definition of the KL divergence.*: Oh, you are right, $\lambda_{i,j}' = 0,1 \implies \varphi_{i,j}=0,1$ but not vice-versa. In this case, then, I believe that using the standard definition of the KL divergence is necessary since the points where $\lambda_{i,j}' \neq 0,1$ but $\varphi_{i,j} \to 0,1$ have $D_{\textnormal{KL}}(\lambda_{i,j}' \lVert \varphi_{i,j}) \to \infty$, which may interfere with the proof of Lemma 3.1. I would like to see how this is resolved in order to verify that the proof is correct.

---

> > > ### Author Response · Authors · 2022-11-15
> > > **Authors' response to follow-up**
> > >
> > > Thank you very much for the additional feedback. We have uploaded a new revision with the suggested changes.
> > >
> > > >Regarding the code and figures: I understand it may require further work, and it seems okay to me to wait for the axis-aligned figures until the final version. Nonetheless, I expect some code (even if only preliminary) before that.
> > >
> > > We have uploaded a preliminary version of the code as supplementary material.
> > >
> > > >Regarding the specifying of Zhang et al. (2020): It seems okay to me, but maybe re-write it as follows: "(e.g., considering a fully connected architecture with sub-differentiable and coordinate-wise Lipschitz activations and weights laying on a compact set)".
> > >
> > > We have rewritten this according to your suggestion in the new revision.
> > >
> > > > Regarding Jin et al. (2022): It improved but is still unclear to me. It mentions that each perturbation may require a linear amount of randomness. Is it possible to show that the overall randomness must be at least linear? You may add a note in the appendix for this purpose if necessary.
> > >
> > > As points are perturbed within a $d$-dimensional ball, indeed, the randomness must be linear in $d$. See the footnote on page 7 in their Arxiv version - https://arxiv.org/pdf/1703.00887.pdf
> > >
> > > We changed "may" to "must" in the new revision.
> > >
> > > > Regarding the definition of the KL divergence [...]
> > >
> > > Our extension is simply the result of Lemma 2.4.
> > >
> > > In Lemma 3.1 we call Lemma 2.4 with parameters $A=B_{i,j-1}, B=X_{i,j-1}, \gamma = b/n_{j-1},\kappa_{A}=\varphi_{i,j}, \kappa_{B}=\lambda_{i,j}'$ and $g(x)=acc(W_{i,j}, x)$.
> > > We explicitly handle the case where $\kappa_{A} = 0,1$ and $\kappa_{B} \neq 0,1 $ in the lemma. We define the extension of $KL_D$ to be consistent with the result of Lemma 2.4.
> > >
> > > Could you please take a look at Lemma 2.4 and let us know if you still believe this extension is problematic?

---

### Official Review · Reviewer_xLLJ · 2022-10-20

**Confidence:** 3
**Correctness:** 3
**Technical Novelty And Significance:** 3
**Empirical Novelty And Significance:** Not applicable
**Recommendation:** 6

**Clarity, Quality, Novelty And Reproducibility:**

Writing of this paper needs improvement. Many mathematical statements seem casual and the structure of paper isn't clear (for example, should there be a formal statement of main theorem in section 3?), let alone typos. The presentation of this paper isn't up to the standards a ML audience expects for this conference.

**Strength And Weaknesses:**

Strength:

The idea of using entropy compression to analyze SGD is novel and interesting.

Weaknesses:

The assumption of accuracy discrepancy isn't natural: unlike other "prior" assumptions, this assumption involves the process of optimization. The assumption is felt to be designed especially for using the entropy compression tool.

The bound $O(\frac{n \log n}{b^2})$ implies $b=\Omega(\sqrt{n})$ because $\rho_i$ is bounded by a constant, likely $\log 2$ in practice for this binary classification problem. I wonder whether a batch size as large as $\Omega(\sqrt{n})$ is practical, or already implies good convergence results itself.

Corollary 3.3 uses a stronger assumption than what's actually needed: it seems to be this only needs to hold on average.

The setting of section 4 seems artificial and I don't see any close relationship to the theme of section 3, except the common tool they use.

**Summary Of The Paper:**

This paper proves that SGD fits the training data whenever the average accuracy discrepancy over epoches, defined as the sum of accuracy improvement over batches, is large enough. Using a similar idea, this paper also proves GD needs a certain amount of randomness to escape local minimas efficiently.

**Summary Of The Review:**

The idea of this paper is very novel and has the potential to be make impacts. However the assumption and result need more clarification/justification and the writing can be improved.

---

> ### Author Response · Authors · 2022-11-09
> **Author's response to Reviewer xLLJ**
>
> Thank you for taking the time to read this paper. Please find our detailed response below.
>
> >The assumption of accuracy discrepancy isn't natural: unlike other "prior" assumptions, this assumption involves the process of optimization. The assumption is felt to be designed especially for using the entropy compression tool.
>
> We think that accuracy discrepancy should be seen as a measure rather than an assumption. We use accuracy discrepancy to provide sufficient conditions for the convergence of SGD based on its dynamics. We show the applicability of this measure in our lower bound in section 4.
>
> >The bound $O(\frac{n \log n}{b^2})$ implies $b = \Omega(\sqrt{n})$  because $\rho_i$ is bounded by a constant, likely $\log 2$ in practice for this binary classification problem. I wonder whether a batch size as large as $\Omega(\sqrt{n})$ is practical, or already implies good convergence results itself.
>
> Apologies, we could not follow your argument here. Could you explain why $\rho_i$ is lower bounded in practice? If it is lower bounded, how do you derive the lower bound for $b$? Why does a lower bound for $\rho_i$ implies a lower bound for $b$ rather than convergences (as we prove in the paper)?
>
> >Corollary 3.3 uses a stronger assumption than what's actually needed: it seems to be this only needs to hold on average.
>
> Thank you for noting this. We've updated this in the new revision.
>
> > The setting of section 4 seems artificial and I don't see any close relationship to the theme of section 3, except the common tool they use.
>
> Section 4 considers the role of randomness in helping GD escape bad local minima. We believe this result sheds some light on why SGD is better than GD in certain scenarios. Furthermore, this question has received some attention in previous work (see "related work" section).
>
> > Many mathematical statements seem casual and the structure of paper isn't clear (for example, should there be a formal statement of main theorem in section 3?),
>
> We've tried to improve the writing in the new revision. Section 3 now has a main theorem.

---

### Official Review · Reviewer_EpmW · 2022-10-27

**Confidence:** 3
**Correctness:** 3
**Technical Novelty And Significance:** 4
**Empirical Novelty And Significance:** Not applicable
**Recommendation:** 6

**Clarity, Quality, Novelty And Reproducibility:**

The main ideas are quite novel end clearly explained. However, the presentation in the technical parts should be improved.

**Strength And Weaknesses:**

Strengths:
- The paper proposes a novel use of a powerful idea, which even though it comes from a different area, it is demonstrated to potentially have interesting applications in optimization and machine learning.
- It is shown that this technique can be used both for positive and negative results.

Weaknesses
- The presentation of the paper could be improved. For example, the results need to be more clearly stated in the introduction and I would also suggest that the main results are stated earlier in each of the sections.


Minor comments:
- Page 5, second bullet: there is a comma missing in the lhs.
- In section 4: I am a bit confused with the use of $m$ and $n$. It seems that they should both mean the size of the dataset. So, the notation needs to be merged.


**Summary Of The Paper:**

This paper is about analyzing the behavior of stochastic gradient descent (SGD) using Kolmogorov complexity.
The authors are using an “entropy compression argument” to prove the termination (and convergence) of SGD under weaker assumptions as well as provide a lower bound on the amount of randomness needed for SGD to escape local minima using perturbations.
The main argument originates from a constructive proof of the Lovasz Local Lemma by Moser, and the idea is that: “Suppose that by looking at the execution of a randomized algorithm, one is able to produce a string shorter than the string of random bits that the algorithm used, albeit it is possible to use that shorter string in order to reconstruct the input random string, then the algorithm should terminate as soon as that happens since it is not possible to compress a string of i.i.d random bits.“
For the first result, the data is split in batches and the algorithm terminates when the accuracy reaches some value<1 (e.g 90%), while for each batch the accuracy is 100%. This allows a more efficient “encoding” of the batch samples since only the subset of correctly classified data could be used. This enables the entropy compression argument to go through and prove termination of the SGD algorithm.
For the second result, it is assumed that the algorithm used $\ell$ uniformly random bits to choose among $2^\ell$ different perturbations. The authors use a similar argument or random string reconstruction and show that a using a sublinear in the size of the dataset number of random bits, it is not possible for SGD to terminate in polynomial time and non-trivial accuracy.




**Summary Of The Review:**

I believe the paper makes a solid conceptual contribution and deserves to be accepted if the writing is improved.

---

> ### Author Response · Authors · 2022-11-09
> **Authors' response to Reviewer EpmW**
>
> Thank you for taking the time to read this paper. We've uploaded a new revision, which hopefully addresses your concerns (minor and major). Please find a detailed response below.
>
> >The presentation of the paper could be improved. For example, the results need to be more clearly stated in the introduction and I would also suggest that the main results are stated earlier in each of the sections.
>
> We tried to improve readability and clarity, following comments from all reviewers. Unfortunately, due to a lack of space, we could not restate the main Theorems in the introduction. We did try to add text to clarify the results. In section 4 we now state the goal of the section directly in the beginning. Any further suggestions regarding writing are welcome.
>
> > In section 4: I am a bit confused with the use of $m$  and $n$ . It seems that they should both mean the size of the dataset. So, the notation needs to be merged.
>
> m is the size of the input to GD, alg 2, which is different than that to Alg 3. That is, Alg 3 calls Alg 2 with a smaller input.

---

### Official Review · Reviewer_GiuD · 2022-10-28

**Confidence:** 2
**Correctness:** 4
**Technical Novelty And Significance:** 3
**Empirical Novelty And Significance:** 2
**Recommendation:** 6

**Clarity, Quality, Novelty And Reproducibility:**

The paper is very novel and generally of good quality. However, the clarity of the paper could be improved by discussing the implications more explicitly. I also point out some typos:
- Throughout the paper: “minimas” should be “minima”
- Pg. 1 footnote, “mean” -> “means”
- Pg. 2: “perturbe”
- Pg. 2, “The randomness we compress are the bits” -> “The randomness we compress is the bits”
- Pg. 3, “Showing that indeed…” is a sentence fragment
- Pg. 3, “pertrubes”
- Pg. 3, by R[0,1] do you mean [0, 1]?
- Pg. 5, “private case of Pinsker’s inequality” what does this mean?
- Pg. 6, “Where in the above…” the word “Where” should not be capitalized; similarly in the paragraph below
- Pg. 8, “Even for extremely…” is a sentence fragment
- Pg. 8, “there must be instances with exponential running time” This is not clear from the paragraph; just because something is not polynomial does not necessarily mean it is exponential.
- Thm. 4.3, “Even if…” is a sentence fragment

**Strength And Weaknesses:**

I found the paper to be very interesting and refreshing, as it adopts a perspective on SGD that I had not considered before. I also appreciate the focus on minimal assumptions, as it seems important to understand the general behavior of SGD.

I believe that the main weakness of the paper is that it is quite abstract, and it is not clear to what extent the findings are applicable to real-world situations. To this extent, more concrete examples in the main text would be appreciated (e.g., are there concrete learning tasks for which the results of the paper are easily illustrated?). Also, it is worth noting that the setting may be too general, and the results of the paper may not be typical for situations encountered in practice.

**Summary Of The Paper:**

This paper studies batch SGD using the theory of Kolmogorov complexity to deduce novel observations about the behavior of SGD. For instance, the paper shows that certain patterns of classification accuracy obtained on the batches would allow the randomness of the batches (the permutation used to select the batches) to be compressed, and this implies a bound on the number of iterations SGD requires before achieving 100% test accuracy on the entire training data set. The paper also considers a lower bound on the amount of randomness needed to escape local minima.

**Summary Of The Review:**

Despite the lack of concrete implications, I like the novelty and I recommend the paper for acceptance. I hope that it will generate interesting follow-up work.

---

> ### Author Response · Authors · 2022-11-09
> **Authors' response to Reviewer GiuD**
>
> Thank you for taking the time to read this paper. We've uploaded a revision which should take care of all the minor comments mentioned. Please find our response to the major issues below.
>
> >I believe that the main weakness of the paper is that it is quite abstract, and it is not clear to what extent the findings are applicable to real-world situations. To this extent, more concrete examples in the main text would be appreciated (e.g., are there concrete learning tasks for which the results of the paper are easily illustrated?). Also, it is worth noting that the setting may be too general, and the results of the paper may not be typical for situations encountered in practice.
>
> Indeed, this is a theory paper, and the results are not immediately applicable. However, we believe that perhaps follow-up work would yield more practical results.
>
> Our upper bound gives a sufficient condition for the convergence of SGD, which might be useful for designing and analyzing new SGD variants. The lower bound highlights the importance of randomness for GD-type algorithms, and might be useful in the design and analysis of new GD variants.

---

### Official Review · Reviewer_w9AC · 2022-10-29

**Confidence:** 4
**Correctness:** 4
**Technical Novelty And Significance:** 2
**Empirical Novelty And Significance:** 2
**Recommendation:** 3

**Clarity, Quality, Novelty And Reproducibility:**

At a technical level, the main theoretical claims of the paper appear sound. As I discussed, though, they either are under restricted assumptions or appear insufficiently motivated in ML practice.

I have a few suggests regarding writing presentation (see above).

**Strength And Weaknesses:**

The paper provides some interesting ideas for studying SGD convergence. To the best of my knowledge, the argument in Section 3 is novel. I think the result is intuitive. That is, if any model training can consistently improve the accuracy on the current batch during SGD, relative to the full dataset, then it is capable of fitting all data. I appreciate the technical part where this idea is formalized.

While the theoretical claim of section 3 is interesting, I have concerns about the setting and motivation. First, the argument hinges upon the assumption that the SGD/GD is run with a sufficiently small step size below $1/L$.  I am not convinced that this holds in practice. In particular, modern neural networks are typically trained with large step sizes at a initial stage, higher than what's suggested by optimization theory, then later annealed as training loss improves. See the literature on "learning rate decay" and "learning rate schedule". This may break the reversibility lemma in section 2 and affect the main claim of section 3.

For section 4, as the author(s) suggested, one motivation is to study whether SGD can escape local minima. It's unclear this is an interesting question (either in theory or practice). In practice, large networks are highly parametrized and capable of fitting the entire train set. And it's observed that standard optimization techniques (SGD and its variants, such as Adam) can converge.
The claim of section 4 seems to be that we need to perturb the input points to avoid getting stuck at a highly sub-optimal local min (with 1/2+$\epsilon$ accuracy). I encourage the author(s) to expand on the conceptual  message here, since as far as I know, this is not required in practical settings.

The main result in sec 3 applies to reshuffling SGD where each epoch uses a fresh random permutation. It's also common in practice that there's a single random permutation (sampled upfront) used by all epochs. (This is called single shuffle SGD; see e.g. Can Single-Shuffle SGD be Better than Reshuffling SGD and GD? by Chulhee Yun, Suvrit Sra, Ali Jadbabaie.) Could the author(s) comment on this setting?

Some parts of the paper can be better written.  In the introduction (especially in the “outline of our techniques” section), it should be clarified what are the random bits that the reversal procedure tries to reconstruct. From early discussion (top of page 2), there are random bits in the initialization of the network, random bits for drawing SGD samples per step, and random perturbations added to the input points. The author(s) should specify what the main target here is.

Finally, the experiments appear somewhat limited, conducted only on shallow neural networks trained with MNIST dataset.

Minor comments
---
1. Page 1: “For every batch, the accuracy of the model after the Gradient Descent (GD) step on the batch is 100%.” When I read it first, it wasn’t clear if this is a claim or a thought experiment. Maybe add: “Suppose hypothetically.”
2. Page 1: “, however” -> “. However”
3. Page 2: “perturbe” -> “perturb”
4. Page 2: The precise termination condition seems confusing here. In one paragraph, it is written that “thus, in our case, termination implies 100% accuracy”. In a later paragraph: “We apply this approach to SGD with an added termination condition when the accuracy over the entire dataset is beyond some threshold. Thus, termination in our case guarantees good accuracy”. Which one is actually applied here?
5. Page 2: “ a sufficiently small GD step” -> “sufficiently small step size”
6. Page 3: “we can sort X by IDs”--- I am not sure this is necessary to say?
7. Page 3: “pertrubes” -> “perturbs”
8. Page 3: “Applying this reasoning to data with random labels we arrive at a contradiction” — I don’t understand this point: what’s the contradiction
9. Section 3: W_ij should be defined explicitly. (I understand it’s clear from the picture.)
10. Section 4: I think it’s better to state the goal of this section at its beginning


**Summary Of The Paper:**

The paper studies the convergence property of SGD using tools from Kolmogorov Complexity.  A key quantity considered here is the accuracy discrepancy---the gap between the model accuracy on a training batch vs the rest of the data set.

The paper provides two results.

* Section 3 shows that during training if the accuracy discrepancy is consistently beyond a global threshold $\beta(n,b)$, depending only on batch size $b$ and data size $n$, then training will converge to full accuracy in constant epochs (namely, after constant cycles over the dataset).
* Section 4 considers randomly perturbing the inputs to allow SGD to terminate with high accuracy. It's shown that SGD may take exponential time unless the amount of randomness used in the perturbations is sufficiently high.

**Summary Of The Review:**

Overall I think the paper has some interesting ideas. However, I am not convinced about the applicability of the claims. The paper can also be improved in terms of clarity and scope of experiments.

---

> ### Author Response · Authors · 2022-11-09
> **Authors' response to reviewer w9AC**
>
> Thank you for taking the time to review this paper. Please find our detailed response to the major issues below, while all minor issues should be resolved in the updated revision.
>
> >First, the argument hinges upon the assumption that the SGD/GD is run with a sufficiently small step size below . I am not convinced that this holds in practice. In particular, modern neural networks are typically trained with large step sizes at a initial stage, higher than what's suggested by optimization theory, then later annealed as training loss improves. See the literature on "learning rate decay" and "learning rate schedule". This may break the reversibility lemma in section 2 and affect the main claim of section 3.
>
> Thank you for relating this result to practical applications of SGD. We would like to note that having a small learning rate is a very common assumption in the theory literature. Compared to other (influential) results we allow quite a large step size. See, for example, "A Convergence Theory for Deep Learning via Over-Parameterization" where the learning rate has an inverse polynomial dependence on both n and L.
>
> Furthermore, our results also hold for the case you describe, although with slightly weaker guarantees. If initially the learning rate is large, but at iteration $t_0$ it goes below our threshold, we can apply our results starting from iteration $t_0$. These results will be a bit weaker as the model in iteration $t_0$ is not a random model, and the parameter $d$ will appear in our bounds.
>
> > For section 4, as the author(s) suggested, one motivation is to study whether SGD can escape local minima. It's unclear this is an interesting question (either in theory or practice). In practice, large networks are highly parametrized and capable of fitting the entire train set. And it's observed that standard optimization techniques (SGD and its variants, such as Adam) can converge. The claim of section 4 seems to be that we need to perturb the input points to avoid getting stuck at a highly sub-optimal local min (with $1/2+\epsilon$ accuracy). I encourage the author(s) to expand on the conceptual message here, since as far as I know, this is not required in practical settings.
>
> The results of section 4 deal with GD. While SGD, and Adam are very efficient at avoiding local minima, they are stochastic methods. What we show in section 4 is that randomness is important to avoid local minima. This provides some explanation as to why stochastic methods (SGD) outperform non-stochastic methods (GD) in certain scenarios.
>
> > It's also common in practice that there's a single random permutation (sampled upfront) used by all epochs. (This is called single shuffle SGD) Could the author(s) comment on this setting?
>
> Thank you for bringing this to our attention. Our current upper bound analysis does not work for this setting, as we require a fresh permutation per epoch. Conceptually, our analysis technique can be applicable if we can show that the more epochs single shuffle SGD executes for, the more random bits from the initial permutations we can reconstruct.
>
> On the lower bound side, we think that the result in section 4 might be relevant here. If we replace Alg 2. by a single shuffle SGD that uses a sublinear number of bits to shuffle the input, then our proof should still go through where the current term for the randomness of the perturbation will be replaced with the bits used for a single shuffle. As this new term is sublinear, the proof should still hold. Therefore, we think that single shuffle SGD must require a linear number of random bits to achieve non-trivial accuracy on the dataset. For example, shuffling only a subset of o(n) will result in poor performance in some instances.
>
> > In the introduction (especially in the “outline of our techniques” section), it should be clarified what are the random bits that the reversal procedure tries to reconstruct. From early discussion (top of page 2), there are random bits in the initialization of the network, random bits for drawing SGD samples per step, and random perturbations added to the input points. The author(s) should specify what the main target here is.
>
> We state in the third paragraph of "outline of our techniques" that "The randomness we compress is the bits required to represent the random permutation of the data at every epoch". Do you mean we should state this sooner? Or is the statement not clear?
>
> > The experiments appear somewhat limited, conducted only on shallow neural networks trained with MNIST dataset.
>
> The main goal of the experiments is to show how the accuracy discrepancy behaves on *random data*, specifically, that it goes below a certain global threshold over time as convergence is impossible. Furthermore, we establish that the dependence of the global threshold is inversely quadratic in the batch size, which is in line with our theoretical results. We are not sure that larger networks \ datasets will strengthen these claims considerably.

---

### Official Review · Reviewer_5jcV · 2022-10-29

**Confidence:** 3
**Correctness:** 3
**Technical Novelty And Significance:** 3
**Empirical Novelty And Significance:** 3
**Recommendation:** 5

**Clarity, Quality, Novelty And Reproducibility:**

Overall, there is room for clarity and quality. Please refer to the section above for comments on clarity and quality. In terms of originality, the paper provides some discussion of SGD and prior literature but does not provide many comparisons against prior work or prior techniques. Some additional discussion / comments in the paper regarding novelty of techniques would be helpful.

**Strength And Weaknesses:**

Overall, the paper is interesting but suffers from clarity issues.

Strengths of the paper:
- The paper discusses SGD, which is an important and relevant topic for the learning community.
- The paper gives many examples for the introduction / motivation
- The usage of Kolmogorov complexity in the analysis of SGD is interesting

Some comments on weaknesses include:
a. It appears that the "reversibility" of Lemma 2.1 and other usages in the paper requires that the smoothness constant L satisfy L>1, but it was not clearly mentioned in the paper. Are there any comments on this and how it affects or puts limitations on the main result?
b. There are parts in the paper that leave a confusing impression. For example, in page 2 "Outline of our techniques" it is stated that "if the accuracy on the entire dataset is 100% we terminate" but immediately at the top of page 3 the paper asks the reader to consider a scenario where "we terminate our algorithm when we achieve 90% accuracy on the entire dataset". This, being in the introduction section, may cause confusion for readers.
c. Following the above point on confusion, there are also other sentences that may require revision. For example, on page 5, what does "private case of Pinsker's inequality" mean? And in the "Representing sets" section, it is said that "some useful bounds" are to be stated there, but there is only one bound Lemma 2.4., is something missing here?
d. Does the result only hold for specific scenarios of SGD? For example, does the results of Section 4 only hold when we need convergence/termination in Alg. 2? What about the scenario when SGD only has one step of GD (such as similar to Algorithm 1)? This seems to show that the results are fairly limited and only suitable for specific scenarios.
e. The main results and flow of the discussion is not very clear and concise. This is also shown through the fact that there are a lot of Lemmas but there is no main Theorem in Section 3 and only one main Theorem at the very end of page 9 in Section 4. The clarity of the paper would be greatly improved with more discussions on the flow of proofs and more interpretations of lemmas.
f. The experiment results were not discussed in detail.
g. Some comments on typos: There are several occurrences in the paper where "perturb" is spelled wrong. Another point is: Is the description of Algorithm 3 missing?

**Summary Of The Paper:**

The paper considers dynamics of stochastic gradient descent (SGD) and relates accuracy with a notion of "accuracy discrepancy". The paper shows that if the "accuracy discrepancy" is large enough, then SGD can find a model with perfect accuracy, while on the other hand if the "accuracy discrepancy" is small, there exists inputs for which a specific gradient descent component within an SGD algorithm terminates in exponential time.

**Summary Of The Review:**

The paper is interesting but has several concerns including those listed in the sections above. Revisions on the paper would help with its clarity and quality.

---

> ### Author Response · Authors · 2022-11-09
> **Authors' comment to Reviewer "5jcV"**
>
> Thank you for taking the time to review this paper. We've uploaded a new revision that hopefully addresses all of your concerns. Please find detailed comments below.
>
> >It appears that the "reversibility" of Lemma 2.1 and other usages in the paper requires that the smoothness constant L satisfy L>1, but it was not clearly mentioned in the paper. Are there any comments on this and how it affects or puts limitations on the main result?
>
> We do not require $L>1$, but only that the learning rate is less than $1/L$. Please let us know if we misunderstood your concern.
>
> >There are parts in the paper that leave a confusing impression. For example, in page 2 "Outline of our techniques" it is stated that "if the accuracy on the entire dataset is 100 we terminate" but immediately at the top of page 3 the paper asks the reader to consider a scenario where "we terminate our algorithm when we achieve 90 accuracy on the entire dataset". This, being in the introduction section, may cause confusion for readers.
>
> This was indeed confusing. We've changed it to be consistent.
>
> >Following the above point on confusion, there are also other sentences that may require revision. For example, on page 5, what does "private case of Pinsker's inequality" mean? And in the "Representing sets" section, it is said that "some useful bounds" are to be stated there, but there is only one bound Lemma 2.4., is something missing here?
>
> The general case of Pinsker's inequality is for two distributions. We use it for the case where the distributions are Bernoulli random variables. We've clarified this in the text. As for the "some useful bounds", it was indeed a typo.
>
> >Does the result only hold for specific scenarios of SGD? For example, does the results of Section 4 only hold when we need convergence/termination in Alg. 2? What about the scenario when SGD only has one step of GD (such as similar to Algorithm 1)? This seems to show that the results are fairly limited and only suitable for specific scenarios.
>
> We're not sure that we fully understand the concern raised here. The results in section 4 are applicable to (non stochastic) gradient descent (Alg. 2), which is quite a general algorithm. To show this result, we introduce Alg. 3 which is an auxiliary algorithm. Essentially the results of section 4 show the importance of randomness for gradient descent convergence and give an indication of why SGD is superior to GD in certain scenarios.
>
> > The main results and flow of the discussion is not very clear and concise. This is also shown through the fact that there are a lot of Lemmas but there is no main Theorem in Section 3 and only one main Theorem at the very end of page 9 in Section 4. The clarity of the paper would be greatly improved with more discussions on the flow of proofs and more interpretations of lemmas.
>
> We tried to improve readability and clarity, following comments from all reviewers. We have changed Lemma 3.2 into a theorem. We would be happy to address any other specific issues regarding the paper's clarity.
>
> > The experiment results were not discussed in detail.
>
> Indeed, due to a lack of space, we had to defer most of the discussion regarding the experiments to the appendix. We tried highlighting the main results in the body of the text in the original submission. Please let us know if there is anything specific that we should add to the main text.
>
> >Some comments on typos: There are several occurrences in the paper where "perturb" is spelled wrong. Another point is: Is the description of Algorithm 3 missing?
>
> Typos should be fixed in the new revision. About Alg. 3, indeed we only say that "To show the above, we define a variant of SGD, which uses GD as a sub procedure (Algorithm 3)." As it only differs from Alg.1 in the termination condition and the call to GD (Alg.2) we assumed it is pretty straightforward. we can expand on more if the reviewer believes it is necessary.
>
> >In terms of originality, the paper provides some discussion of SGD and prior literature but does not provide many comparisons against prior work or prior techniques. Some additional discussion / comments in the paper regarding novelty of techniques would be helpful.
>
> To the best of our knowledge, this is the first use of entropy compression in this field. We did our best to compare to previous work and would happily address any additional related works the reviewer has in mind.

---

> > ### Comment · Reviewer_5jcV · 2022-11-24
> > **Additional Comments**
> >
> > Thanks for the responses to my questions.
> >
> > Apologies for misunderstandings in the first comment; my comment was similar in spirit to the point raised by reviewer w9AC on the effects of step size to important assumptions such as the reversibility lemma. I have read the discussions and replies on that review and have no additional questions relating to this.
> >
> > The concern raised for "specific scenarios of SGD" is quite similar to the first weakness point raised by reviewer T4KT, and I was intending to ask if there are algorithms, results or insights on possible results that do not require termination of the algorithm given in the paper, or if there are weaker results that may be obtained if the goal is not to prove convergence of SGD but something else. Apart from comments given to reviewer T4KT, are there any comments on extensions in this direction?
> >
> > I agree with other reviewers on comments on clarity and suggest detailed revisions be made in following versions of the paper.

---

> > > ### Author Response · Authors · 2022-11-24
> > > **Response to additional comments**
> > >
> > > Thank you for the additional comments.
> > >
> > > >The concern raised for "specific scenarios of SGD" is quite similar to the first weakness point raised by reviewer T4KT, and I was intending to ask if there are algorithms, results or insights on possible results that do not require termination of the algorithm given in the paper, or if there are weaker results that may be obtained if the goal is not to prove convergence of SGD but something else. Apart from comments given to reviewer T4KT, are there any comments on extensions in this direction?
> > >
> > > Perhaps we misunderstood what you meant by "require termination", but we would like to emphasize that non of our results requires any algorithm to terminate. The termination conditions are only added for the sake of analysis. For the upper bound (section 3), we provide a sufficient condition for the convergence of SGD. In section 4, we provide lower bounds for GD when the amount of randomness is limited. Both bounds hold for the vanilla versions of the algorithms without any added stopping conditions.
> > >
> > > The first point raised by reviewer T4KT seems to be regarding the fact that we only provide sufficient conditions for convergence rather than actually proving convergence. Nevertheless, we believe that our results provide a new approach to attacking the problem of SGD convergence. Furthermore, the fact that we used our framework to derive lower bounds for GD with bounded randomness implies that it is indeed useful beyond "just" providing a sufficient condition for SGD convergence. We believe our lower bound can also be applied to other algorithms, for example, single shuffle SGD with limited randomness. Please see our reply to reviewer w9AC for more details.

---

### Official Review · Reviewer_dQLw · 2022-10-31

**Confidence:** 4
**Correctness:** 4
**Technical Novelty And Significance:** 4
**Empirical Novelty And Significance:** Not applicable
**Recommendation:** 8

**Clarity, Quality, Novelty And Reproducibility:**

Minor issues:
-- Theorem 4.3: I don’t understand what ½ + θ(1) means
-- Instead of $\tilde{\nabla f_{A}}$ (and others), I would write $\tilde{\nabla f}_{A}$. The long tilde looks really weird
-- I think “w.h.p.” is commonly defined as 1 - n^-c for an arbitrary large c
-- Page 6:” Where in the above we used Stirling’s approximation” and “Where the efficiency of reconstruction is expressed via ρ_i”: should start commas (instead of being new sentences).
-- I think the relation between f and acc was never mentioned


**Strength And Weaknesses:**

I have the following concerns about the paper:

-- As I understand, the discussion near Theorem 2.2 is really important, since it shows that the provess is invertible. Moreover, the algorithm actually relies on the fact that the set of possible values is finite, to find the original point in finite time.

However, the conditions in Theorem 2.2 don’t necessarily hold in practice. To give an example, for f(x)=x^2 / 2, with the step size 1e-16, for the double type, the gradient step starting from points 1e16 + 2 and 1e16 produce the same result.

While I understand that the example is artificial, it suffices to show that the process is, in general, not invertible (when working in doubles). I’m not sure, but it might be important for the rest of the paper, since you do rely on the invertibility. I think that it should be clarified why the conditions of Theorem 2.2 actually hold.

-- “It is clear that w.h.p we cannot train a model with d parameters that achieves any accuracy better than 1/2 + o(1) on X” - I’m not sure this statement is true. You probably should use non-random labels and counting argument.



**Summary Of The Paper:**

The paper considers studies dynamics of mini-batch SGD using Kolmogorov complexity. They define a notion of accuracy discrepancy as a KL-divergence between accuracy of the model on previous batches in the current epoch and on the last batch. Using the fact that the random strings used for generating the epoch’s permutation are incompressible, they bound the accuracy discrepancy throughout the algorithm execution.


**Summary Of The Review:**

Solid paper, accept.

---

> ### Author Response · Authors · 2022-11-09
> **Authors' response to Reviewer "dQLw"**
>
> Thank you very much for taking the time to review this paper. We have uploaded an updated revision where we attempted to address all issues raised (major and minor). Please find our detailed response to the issues raised below.
>
> >As I understand, the discussion near Theorem 2.2 is really important, since it shows that the provess is invertible. Moreover, the algorithm actually relies on the fact that the set of possible values is finite, to find the original point in finite time.
> However, the conditions in Theorem 2.2 don’t necessarily hold in practice. To give an example, for f(x)=x^2 / 2, with the step size 1e-16, for the double type, the gradient step starting from points 1e16 + 2 and 1e16 produce the same result.
>
> This is indeed a valid point, we must assume numerical stability of the function for our analysis to hold. It is quite common to ignore numerical stability issues in the literature, and we believe that other known convergence bound will also encounter problems if numerical stability is taken into account.
> Nevertheless, our analysis is still robust in the following ways:
> - It is sufficient that the function is numerically stable only along the path that SGD traverses (and not all possible points).
> - Our analysis is robust to "some" numerical instability. That is, if the number of possible points leading to the same point for every iteration is bounded by $z$ our analysis still goes through with an additional cost of $\log z$ for reconstruction. This will change the value of $\beta(n,b)$ if $z$ is sufficiently large.
>
> We believe that formally proving convergence under numerical instability can be a natural direction for future work. It would be especially interesting to quantify how the precision affects the convergence guarantee.
>
> >“It is clear that w.h.p we cannot train a model with d parameters that achieves any accuracy better than 1/2 + o(1) on X” - I’m not sure this statement is true. You probably should use non-random labels and counting argument.
>
> Thank you for pointing this out. This is indeed not a trivial claim as we initially thought. However, it is true. We've added a formal proof of the statement using the incompressibility of the random labels (Lemma A.4 in the appendix).
>
> ## Minor issues
> We have tried to address all of the issues raised in the revision. Below we answer what we can, while other issues (tilde over f and grammatical issues) should be resolved in the new revision.
>
> >Theorem 4.3: I don’t understand what ½ + θ(1) means
>
> It means it is impossible to get an approximation better than $1/2 + c$ where $c\leq 1/2$ is a constant that does not depend on $n$. We used asymptotic notation for simplicity of notation.
>
> >I think “w.h.p.” is commonly defined as 1 - n^-c for an arbitrary large c.
>
> This is probably the most common definition, however, other definitions are also often used (for example, reviewer T4KT uses a different definition). Our results hold for this definition as well. We wrote it as we did to avoid introducing further notation.
>
> > I think the relation between f and acc was never mentioned
>
> While in practice you will probably derive acc from f (by rounding etc.), we do not assume any direct relation except that both are computed using W. That is, we make no assumption on how elements are classified using the model.
>
> The properties of f guarantee that the process is reversible, and the properties of acc govern the behavior of the accuracy discrepancy. We believe that using a degenerate accuracy function (e.g., classifying all elements identically or randomly using W) will simply result in very low accuracy discrepancy.
>
> An interesting future line of work would be to connect the two. That is, showing that executing SGD on a specific loss function (from which acc is derived) results in accuracy discrepancy above some threshold, which in turn must lead to convergence.

---

### Decision · Program_Chairs · 2023-01-20

**Decision:**

Reject

**Justification For Why Not Higher Score:**

There are two major reasons that I recommend rejection:

- The main result relates the convergence of SGD to the accuracy discrepancy, which, in turn, relies on the optimization process, as discussed by Reviewer w9AC,  Reviewer xLLJ and Reviewer T4KT. It establishes the connection between the convergence of SGD and the accuracy discrepancy which we may not have a good understanding of, rather than something that we have a good understanding of. The main result does not solve the problem that it claims to solve. Given the assumption about the accuracy discrepancy, the proof is straightforward and thus lacks novelty.
- The presentation of this paper needs improvement.

**Justification For Why Not Lower Score:**

N/A

**Metareview: Summary, Strengths And Weaknesses:**

Summary
---
This paper studies the convergence property of stochastic gradient descent (SGD) using tools from Kolmogorov complexity. The authors define a quantity called the "accuracy discrepancy," which is the gap between the model accuracy on a training batch versus the rest of the dataset. They show that if the accuracy discrepancy is consistently beyond a certain threshold, then training will converge to full accuracy in a constant number of epochs. They also consider the use of randomly perturbed inputs to allow SGD to terminate with high accuracy, and show that SGD may take exponential time unless the amount of randomness used in the perturbations is sufficiently high. The authors use these results to provide a lower bound on the amount of randomness needed for SGD to escape local minima using perturbations. Overall, the paper aims to provide insights into the behavior of SGD and its convergence properties using the theory of Kolmogorov complexity.

Strengths:
---
- The paper provides some interesting ideas for studying SGD convergence, which is an important and relevant topic for the learning community.
- The paper provides examples for the introduction/motivation, and uses tools from Kolmogorov complexity to study SGD.

Weaknesses
---
- The main statement of the convergence of SGD in the paper provides a sufficient metric to determine when the algorithm will converge, but it does not give a convergence result for SGD itself. To determine when SGD will converge, one must run the algorithm and calculate the accuracy discrepancies. Therefore it would be helpful if the paper acknowledged this and positioned its contribution as a finding of a sufficient condition for SGD convergence, rather than a direct convergence result for SGD.
- Multiple reviewers complained about the presentation of this paper.



**Summary Of Ac-Reviewer Meeting:**

Meeting Notes:

- The meeting was held to discuss ICLR'23 submission 6217
- Attendees included Reviewer w9AC, Reviewer 5jcv, Reviewer GiuD, and the area chair

Reviewer w9AC provided the following feedback:

- The theory presented in the paper relies on strong and arguably artificial assumptions that provide little insight into the behavior of the method in practice. Additionally, the proof techniques are not particularly novel or significant. Overall, w9AC does not believe the paper has much potential for further impact. This point was also discussed in the reviews of  Reviewer xLLJ and Reviewer T4KT. This paper relates the convergence of SGD to the accuracy discrepancy, which relies on the optimization process. This limits the usability and importance of the result.
- The paper makes artificial assumptions about the step size being smaller than $1 / L$ in order to achieve recovery.
- The method may not be generalizable to other variants of SGD, as the method requires sampling a fresh random permutation every epoch.

Reviewer 5jcv noted that the presentation of the paper needs improvement.

Reviewer GiuD agreed with the comments made by Reviewers w9AC and 5jcv and added that the paper was not presented clearly enough.